# EASTR: Identifying and eliminating systematic alignment errors in multi-exon genes

Ida Shinder [1,2] ✉, Richard Hu [2,3], Hyun Joo Ji [2,3], Kuan-Hao Chao [2,3] & Mihaela Pertea [2,3,4,5] ✉

Accurate alignment of transcribed RNA to reference genomes is a critical step in the analysis of gene expression, which in turn has broad applications in biomedical research and in the basic sciences. We reveal that widely used splice-aware aligners, such as STAR and HISAT2, can introduce erroneous spliced alignments between repeated sequences, leading to the inclusion of falsely spliced transcripts in RNA-seq experiments. In some cases, the 'phantom' introns resulting from these errors make their way into widely-used genome annotation databases. To address this issue, we present EASTR (Emending Alignments of Spliced Transcript Reads), a software tool that detects and removes falsely spliced alignments or transcripts from alignment and annotation files. EASTR improves the accuracy of spliced alignments across diverse species, including human, maize, and *Arabidopsis thaliana*, by detecting sequence similarity between intron-flanking regions. We demonstrate that applying EASTR before transcript assembly substantially reduces false positive introns, exons, and transcripts, improving the overall accuracy of assembled transcripts. Additionally, we show that EASTR's application to reference annotation databases can detect and correct likely cases of mis-annotated transcripts.

RNA sequencing (RNA-seq) is a widely used method for quantifying gene expression and characterizing transcriptome diversity. However, repetitive sequences can in some circumstances induce splice-aware aligners, such as STAR and HISAT2, to create spurious introns spanning two nearby repeats. Repeat elements constitute a significant portion of many genomes, comprising 21% of the *Arabidopsis thaliana* genome[1], 53% of the human genome[2], and 85% of the *Zea mays* genome[3]. Repeated elements frequently occur in close proximity; for example, *Alu* elements, a primate-specific transposable element (TE), appear in over a million copies in the human genome, with an average frequency of once every 3000 bases[4]. The close proximity of repeat elements complicates distinguishing spliced and contiguous alignments, particularly in tissues and organisms with high TE expression.

Repeat elements pose challenges not only due to their proximity but also due to their high degree of polymorphism. The variations among individuals and between loci can confound computational methods attempting to distinguish correct and incorrect spliced alignments. As we will show below, this can result in the inclusion of transcripts with spurious junctions in human gene catalogs such as CHESS[5], RefSeq[6], and GENCODE[7].

Computational methods face inherent limitations when aligning short reads from sequencing data to genomes with numerous repeat elements. Read lengths are often shorter than the full length of

[1]Cross Disciplinary Graduate Program in Biomedical Sciences, Johns Hopkins School of Medicine, Baltimore, MD, USA. [2]Center for Computational Biology, Johns Hopkins University, Baltimore, MD, USA. [3]Department of Computer Science, Johns Hopkins University, Baltimore, MD, USA. [4]Department of Biomedical Engineering, Johns Hopkins School of Medicine and Whiting School of Engineering, Baltimore, MD, USA. [5]Department of Genetic Medicine, Johns Hopkins School of Medicine, Baltimore, MD, USA. ✉e-mail: ishinde1@jhmi.edu; mpertea@jhu.edu

repeated sequences, complicating the identification of the true origin of multi-mapped reads. Pseudogenes also present challenges during alignment, as reads that should be aligned across a splice site at their original location may be aligned end-to-end to a pseudogene copy. HISAT[8] addresses this by prioritizing spliced alignments over contiguous alignments if the spliced alignment has no mismatches compared to the contiguous alignment (Fig. 1). If the two alignments are equally good, HISAT2 will report both alignments. However, this approach can lead to misalignments of other types of repeats such as transcripts with variable numbers of tandem repeats (VNTRs) or TEs that have accumulated variation and diverged in sequence over evolutionary time.

In this work, we introduce EASTR (Emending Alignments of Spliced Transcript Reads), a computation tool designed to address the issue of incorrect spliced alignment associated with repeated sequences. EASTR effectively detects and removes erroneous spliced alignments by examining the sequence similarity between the flanking upstream and downstream regions of an intron and the frequency of sequence occurrence in the reference genome. Our results indicate that EASTR substantially improves the quality of spliced alignments across a wide range of species, thereby enhancing the reliability of downstream analyses that rely on accurate spliced alignment, such as transcript assembly. In addition, our findings show that EASTR is effective at detecting inaccuracies in existing reference annotations.

## Results

EASTR employs a multi-step strategy to identify and eliminate spurious splice junctions, focusing on the similarity of sequences flanking a given splice junction as well as the occurrence frequency of these sequences within the reference genome. The algorithm's workflow is visually outlined in Fig. 2, with further details provided in the Methods section. As depicted in Fig. 2, the first step in EASTR's workflow is to extract the sequences flanking a target splice junction and assess their similarity with alignment. When sequence similarity between flanking sequences is observed, a subsequent analysis is conducted to assess the frequency of these sequences within the genome.

The similarity between flanking sequences can have multiple origins, from local elements like tandem repeats to more global features like *Alu* elements. Rather than differentiating between these global and local origins of repetitive elements, EASTR categorizes alignments as either "two-anchor" or "one-anchor" as depicted in Fig. 2 (panels B and

C, respectively). In a two-anchor alignment, significant sequence similarity between both flanking regions allows for potential splice alignment from either end of the artifactual "junction". One-anchor alignments may occur under different scenarios: for example, they could result from a repeat sequence limited to exonic regions, often due to duplicated exons, or they may stem from variations in the number and sequence of tandem repeats where an intronic region of one flanking sequence aligns to the "exonic" region of its counterpart. This can occur due to polymorphisms between a reference genome and a specific sample. EASTR considers the frequency of duplicated sequences in these scenarios. Additionally, EASTR also assesses the uniqueness of the hybrid sequence formed by concatenating the 5' and 3' exons to eliminate the possibility of misclassifying duplicated exons as spurious. Generally, a splice junction is deemed spurious if its flanking sequences align to each other and map to multiple genomic locations, or if a hybrid sequence exists elsewhere in the genome. Alignments supporting these spurious junctions can be selectively removed to generate a refined alignment file. Further technical details are provided in the Methods section.

To demonstrate the versatility and applicability of EASTR, we applied it to three model organisms with distinct genomic repeat content and to samples with different library preparation methods. The organisms and tissues we selected were human brain[9], *Zea mays* (maize) leaves[10] and pollen[11], and *Arabidopsis thaliana*[12]. Our objective was to highlight the effectiveness of EASTR in enhancing alignment accuracy across a broad range of organisms and experimental designs.

To evaluate the impact of EASTR alignment filtering on downstream analyses, we assessed the accuracy of transcript assembly before and after filtering the alignments. We used StringTie2[13] to assemble transcripts from both HISAT2[14] and STAR[15] alignments, as well as from the EASTR-filtered alignments. Our results described below demonstrate that filtering with EASTR prior to assembly improved both sensitivity and precision, and reduced the number of non-reference introns, exons, and transcripts, which are more likely to represent transcriptional noise[16].

In addition to filtering alignment files, we also used EASTR to identify potentially erroneous transcripts in reference annotation databases for each of the selected model organisms. By examining the sequence similarity between the flanking upstream and downstream regions of introns, EASTR was able to detect transcripts in the annotation that may have been incorrectly annotated due to spliced alignment errors between repeat elements.

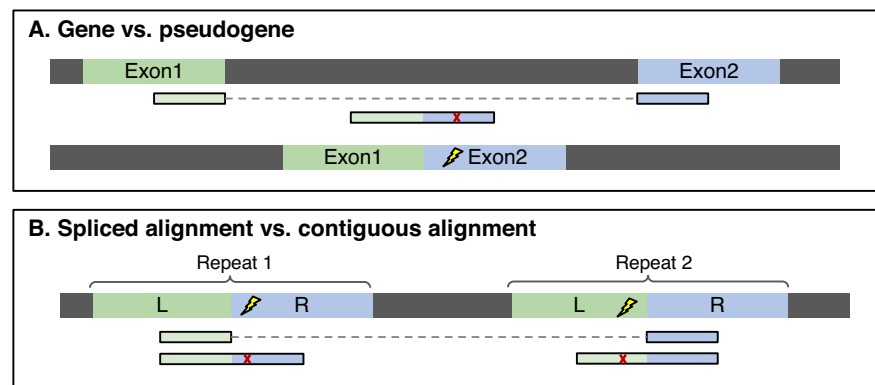

**Fig. 1 | Overview of HISAT2 algorithm's performance in two alignment scenarios. A** The correct spliced alignment without mismatches at a gene locus is favored over an unspliced alignment with one mismatch to a pseudogene. HISAT2 accurately aligns a read (blue and green rectangles) originating from a gene with exon 1 (green) and exon 2 (blue), which also aligns contiguously to a processed pseudogene on a different chromosome. The read contains a single mismatch (marked with an x) to the pseudogene. When the read is aligned to its correct location, spanning an intron, the alignment has zero mismatches. **B** An incorrect spliced alignment caused by consecutive repeats can occur when a read (blue and green rectangles) originates from either repeat 1 or repeat 2, but due to either a base-calling error or a polymorphism, it contains mismatches (marked with x) to both repeat elements. The correct alignment is end-to-end, spanning an entire repeat. Because the repeat's left arm (marked L) ends with AG and the right arm starts with GT (introducing a sequence resembling a canonical splice site), the aligner can generate a false intron and align the read without mismatches.

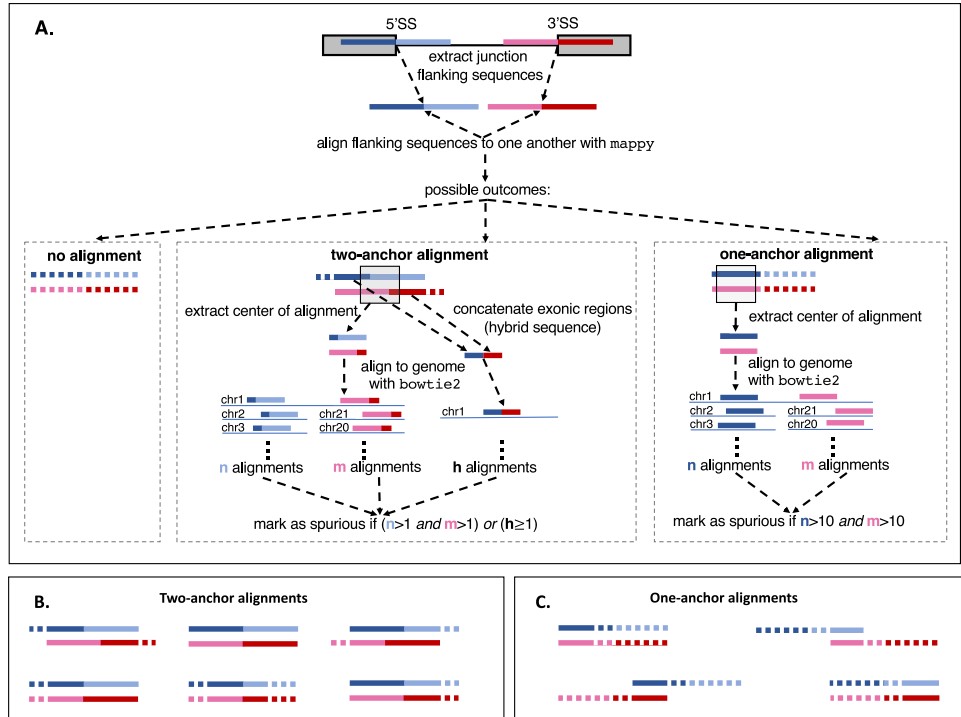

**Fig. 2 | EASTR algorithm for identifying spurious spliced alignments. A** EASTR initiates by extracting sequences centered on the 5′ and 3′ splice sites of a given junction, extending equidistantly from each splice site (SS). The sequence encompassing the 5′SS is represented in blue, while the sequence encompassing the 3′ SS is depicted in red. Deeper hues of blue and red, referred to as exon overhangs (default: 50 bp), denote segments originating from "exonic" regions. These sequences are aligned against each other using `mappy`[26], yielding one of three possible outcomes: an absence of alignment (indicated by dashed lines throughout the figure), a two-anchor alignment where both the upstream and downstream sequences meet specific start and end position criteria within the exon overhangs, allowing for the potential splice-aligning of a read originating from either end of the junction, or a one-anchor alignment for all other instances. In cases involving the latter two outcomes, the alignment is deemed spurious if a defined central region (default:30 bp, marked by a gray square) within the aligned flanking sequences maps to multiple genomic locations as identified by `bowtie2`[25] (more than once for two-anchor alignment or over 10 times for one-anchor alignments). Moreover, a two-anchor alignment is also designated as spurious if a hybrid sequence formed by concatenating the two exonic regions neighboring the two splice sites (5′SS and 3′SS) is present elsewhere in the genome. **B** Examples of two-anchor alignments. **C** Examples of one-anchor alignments.

## Human

We evaluated EASTR's performance on paired RNA-seq datasets from developing and mature human dorsolateral prefrontal cortex (DLPFC). The datasets were obtained using both poly(A) selection and rRNA-depletion (ribo-minus) library preparation methods from cytoplasmic and nuclear fractions from three prenatal and three adult samples[9].

Focusing initially on alignment accuracy, we applied EASTR to identify putative erroneous junctions in the alignment files of 23 DLPFC samples. Our analysis revealed that the vast majority of the alignments flagged for removal by EASTR corresponded to junctions that were not present in the RefSeq reference annotation. On average, EASTR removed 3.4% (5,208,893/ 153,192,435) and 2.7% (3,599,371/ 134,202,142) of all HISAT2 and STAR spliced alignments, respectively. Of the removed alignments, only 0.2% (9,114) in HISAT2 and 0.3% (9,101) in STAR supported 114 and 119 reference-matching junctions, respectively. EASTR marked these as erroneous in the RefSeq reference annotation, and this small subset is discussed in more detail later. Nearly all of the alignments targeted for removal by EASTR were at non-reference junctions: 99.8% (5,199,779) in HISAT2 and 99.7% (3,590,270) in STAR, corresponding to 138,111 and 75,273 non-reference junctions, respectively. This reduction in the number of non-reference junctions was consistent across all 23 samples. More details are provided in Supplementary Data 1 and Fig. 3.

Having demonstrated EASTR's effectiveness in identifying spurious alignments, we further examined the influence of library preparation methods. We found that the ribo-minus library method had a higher proportion of spuriously spliced alignments compared to the poly(A) selection method. Of the 23 samples, 11 pairs were processed using both library selection methods. In ribo-minus samples, EASTR flagged 8.0% (4,145,349/51,742,668) and 6.4% (2,481,034/39,030,763) of HISAT2 and STAR alignments as erroneous, respectively, compared to only 1.0% (1,063,544/101,449,767) and 1.2% (1,118,337/95,171,379) in poly(A) samples. Furthermore, our findings suggest that the developmental stage may be a relevant factor to consider during alignment and downstream RNA-seq analysis. Comparing ribo-minus adult to ribo-minus neonatal samples revealed that, in general, prenatal samples had a higher rate of removed spliced alignments in comparison to adult samples (Supplementary Data 1).

Next, we investigated EASTR's ability to differentiate between genuine and erroneous splicing events in the context of transcribed repetitive regions, such as human endogenous retroviruses (HERVs). Identifying spurious splicing events within transcribed repetitive regions, such as human endogenous retroviruses (HERVs), presents a complex challenge due to the abundance of highly similar copies that frequently result in multiple alignments or spuriously spliced alignments. However, not all spliced alignments in these regions are erroneous, emphasizing the need for EASTR to carefully discern between valid and spurious splice sites. To assess EASTR's robustness in differentiating between genuine and spurious junctions in this context, we used SpliceAI[17], a machine learning-based splice site prediction tool, to score splice junctions that overlap HERV elements in the previously analyzed 23 DLPFC samples. We hypothesized that EASTR would predominantly retain junctions with higher SpliceAI scores, filtering out those with notably lower scores. Of the 1,179 HERV-to-HERV

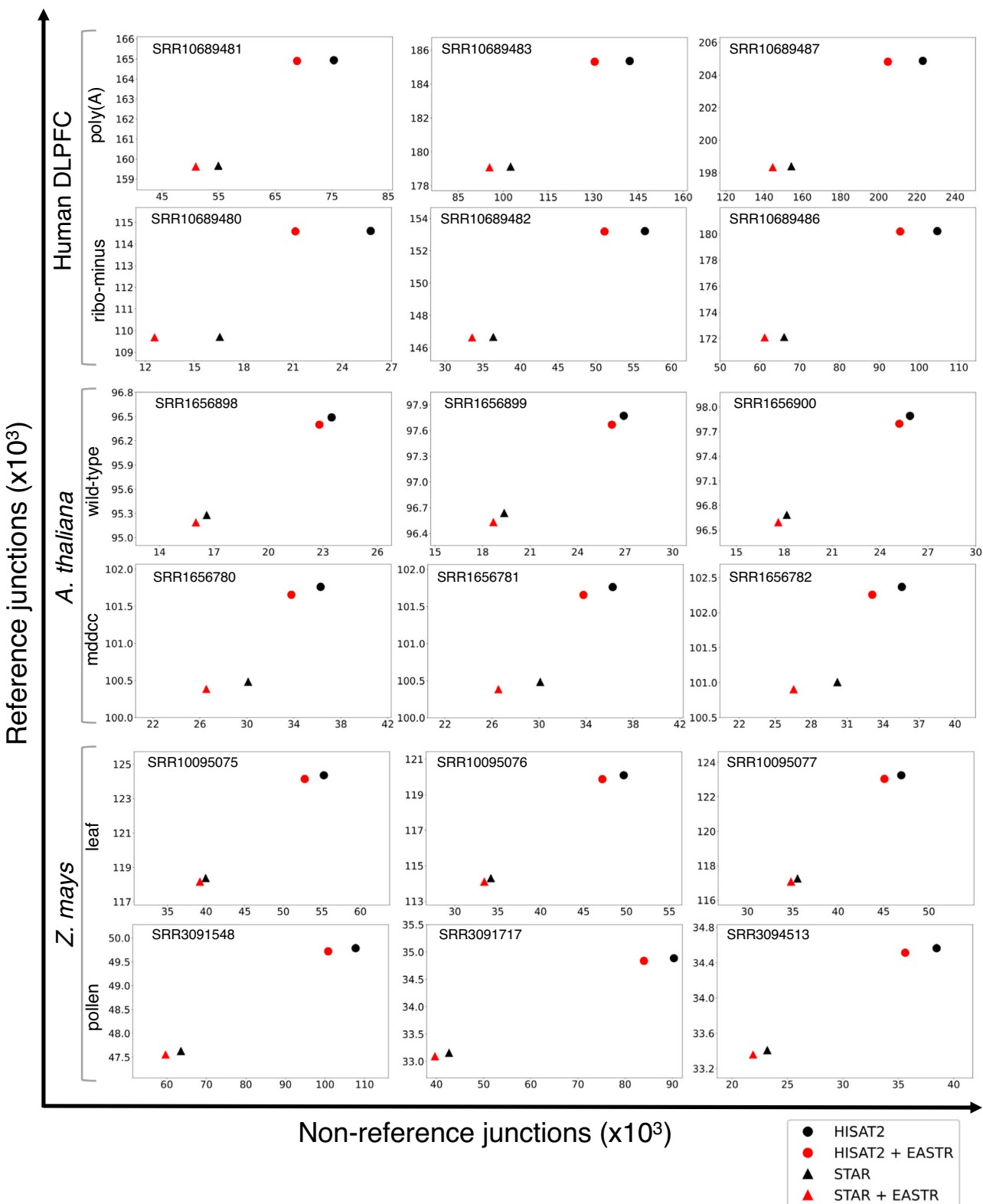

**Fig. 3 | Effectiveness of EASTR filtering in distinguishing between reference and non-reference junctions across various sample types.** We chose three samples from each dataset, including human DLPFC polyA and ribo-minus, *A. thaliana* wild-type and *mddcc* strains, and *Z. mays* lower leaf and mature pollen. The *y*-axis indicates the count of junctions matching the reference annotation for a given sample, whereas the *x*-axis shows the count of junctions not present in the reference annotation.

**Table 1 | Impact of EASTR filtering on transcriptome assembly metrics**

| Aligner | Species | Group | Non-reference introns %Δ | Non-reference exons %Δ | Non-reference transcripts Δ | Reference transcripts Δ |
|---|---|---|---|---|---|---|
| HISAT2 | Human DLPFC | ribo-minus | −5.1% to −22.1% | −3.7% to −19.8% | −148 to −1,076 | +0 to +52 |
| | | polyA | −6.1% to −9.1% | −4.3% to −7.7% | −199 to −797 | +3 to +135 |
| | *A. thaliana* | mddcc | −3.6% to −7.4% | −7.1% to −8.2% | −119 to −176 | −7 to −11 |
| | | wild-type | −5.0% to −5.8% | −7.2% to −9.3% | −86 to −94 | −8 to −14 |
| | *Z. mays* | leaf | −2.1% to −3.5% | −2.5% to −3.2% | −232 to −264 | −23 to −43 |
| | | pollen | −3.5% to −6.2% | −3.8% to −5.4% | −237 to −456 | +5 to +12 |
| STAR | Human DLPFC | ribo-minus | −3.7% to −14.3% | −2.3% to −12.9% | −89 to −603 | −2 to +42 |
| | | polyA | −5.3% to −7.7% | −3.2% to −6.0% | −162 to −561 | +7 to +86 |
| | *A. thaliana* | mddcc | −2.7% to −3.2% | −5.6% to −6.0% | −87 to −100 | −5 to −13 |
| | | wild-type | −4.9% to −6.3% | −8.1% to −8.9% | −56 to −68 | −11 to −19 |
| | *Z. mays* | leaf | −1.0% to −1.3% | −1.2% to −1.8% | −135 to −185 | −32 to −35 |
| | | pollen | −1.2% to −2.1% | −1.3% to −1.9% | −65 to −140 | +2 to +10 |

Assemblies generated from unfiltered alignments to those filtered with EASTR across different species and conditions. The table shows the percentage change (%Δ) in the number of non-reference introns and non-reference exons, and the change in count (Δ) of non-reference and reference transcripts.

junctions identified by either HISAT or STAR, EASTR removed 375 junctions. Consistent with our hypothesis, these removed junctions had SpliceAI scores 6 to 19 times lower than the scores of the retained junctions (further details are provided in Supplementary Note 1; SpliceAI scores for the EASTR flagged and retained HERV junctions can be found in Supplementary Data 16 and 17, respectively).

To assess the impact on transcript assembly, we utilized StringTie2 to assemble transcripts from both unfiltered and EASTR-filtered HISAT2 and STAR alignments. We compared the resulting transcript assemblies to the RefSeq human reference annotation and found that filtering with EASTR improved transcript assembly quality, as summarized in Table 1. The decrease in the number of non-reference introns and exons, as well as the relative improvement in transcript-level precision, can be observed across all samples and alignment tools (Supplementary Data 4 and 7). Importantly, aligning with HISAT2 and subsequently filtering with EASTR did not compromise transcript-level sensitivity in any sample or experimental condition and even resulted in slight improvements relative to the reference annotation. When aligning with STAR and then filtering with EASTR, the decline in transcript-level sensitivity was almost zero, affecting only 1–2 transcripts in two samples. Our results strongly support the adoption of EASTR to enhance transcriptome assembly precision while preserving or improving sensitivity.

Further, we applied EASTR to scrutinize the reliability of junctions in widely used human reference annotation catalogs, including RefSeq (version 110), CHESS (version 3.0), GENCODE (version 41), and MANE (version 1.0)[18]. EASTR detected 365 potentially spurious introns across 581 transcripts and 237 genes in RefSeq, 192 introns across 319 transcripts and 124 genes in CHESS, and 411 introns across 475 transcripts and 344 genes in GENCODE (Supplementary Data 10–12). Notably, we also identified one incorrect MANE transcript, as discussed below.

Narrowing our focus to specific gene families with complex genomic structures, we chose the primate-specific NBPF family as a case study for dissecting the challenges associated with accurate splice site annotation. Our findings reveal that gene families characterized by frequent gene duplication, complex repetitive structures, and variable copy numbers across individuals and populations are particularly susceptible to splicing errors in reference annotations. To illustrate, the NBPF family harbors a two-exon repeat unit known as Olduvai that has expanded through tandem duplication[19,20]. Our analysis centered on the *NBPF20* gene, one of the longest members of this family with a remarkable expansion of the Olduvai repeat unit[19]. As shown in Fig. 4A, the presence of the two-exon repeat unit in the *NBPF20* gene presents challenges in accurately distinguishing between contiguous and

spliced alignments. The RefSeq transcript NM_001278267 (CHESS transcript CHS.2819.2), displays sequence homology between exons 125_1 and 126_1, as well as the "intronic" region between them, suggesting they form two halves of a repeated exon. A comparison of shortened exons 125_1 and 126_1 in transcript NM_001278267 with their full-length counterparts in exons 124_2 and 126_2 in transcript NM_001397211 supports this assertion. As a result, spliced alignments supporting exons 125_1 and 126_1 also align contiguously to exons 124_2 or 126_2. Additionally, a sharp drop in coverage of spliced alignments drops abruptly at the break in sequence homology between the two paralogous exons, further indicating inaccurately spliced alignments (Fig. 4A, coverage and alignment tracks).

Further extending our analysis, we examined the characteristics of introns that EASTR flagged as spurious. A notable fraction of flagged introns were less than 100 bp in length (36% in CHESS, 45% in RefSeq, and 11% in GENCODE), coinciding with regions containing a variable number of tandem repeat (VNTR) polymorphisms. For example, the *PER3* gene contains a VNTR with either 4 or 5 repeated 54 bp sequences[21]. In RefSeq transcript NM_001289861 (CHESS3 CHS.278.18, GENCODE ENST00000614998.4), EASTR identified a 54 bp intron that matches the periodicity of the VNTR region (Fig. 4B). Only 4 out of 27 alignments supporting this intron are without mismatches to the reference genome and have sufficient overhangs extending beyond the region of homology on both ends of the junction. These alignments are likely indicative of an indel, rather than an intron. The remaining 23 alignments exhibit either short overhangs on one end of the junction not extending beyond the homologous region or contain mismatches to the reference, or a combination of both (Fig. 4B, alignment track).

Lastly, in our examination of the MANE catalog, a recently established database that selects one isoform for each protein-coding gene to serve as the representative transcript for that gene, and on which RefSeq and GENCODE agree perfectly, we identified a *TCEANC* gene transcript (RefSeq: NM_001297563, CHESS: CHS.57562.1, GENCODE: ENST00000696128) containing an intron that appears erroneous (Supplementary Fig. 1 and Supplementary Data 13). As further detailed in Supplementary Note 2, this intron features splicing between two consecutive *Alu* elements sharing 84% sequence identity, potentially causing alignment ambiguity that may be compounded by individual polymorphisms. Moreover, the inclusion of the second *Alu* element disrupts the open reading frame (ORF), shortening it substantially (Supplementary Fig. 1). The questionable intron is further characterized by a low-quality splice site acceptor motif, as depicted in Supplementary Fig. 2. Other gene catalogs exhibit many additional *Alu-Alu*

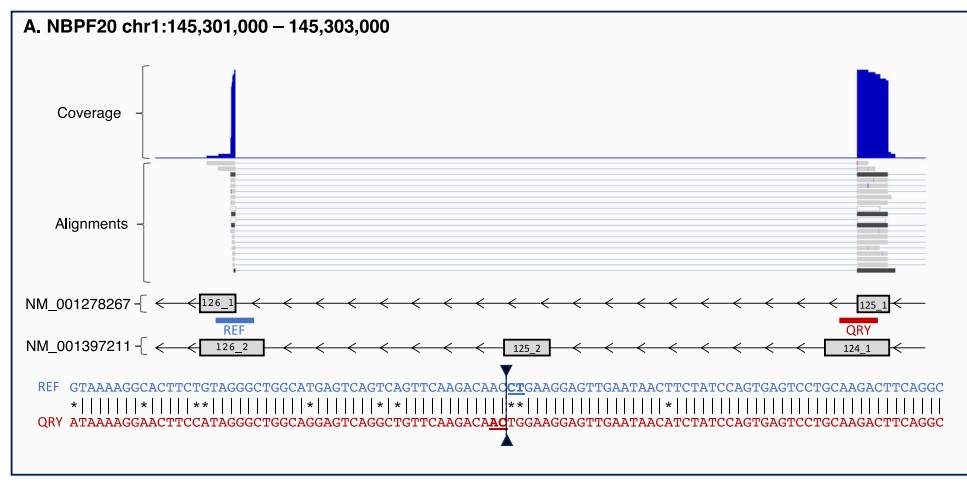

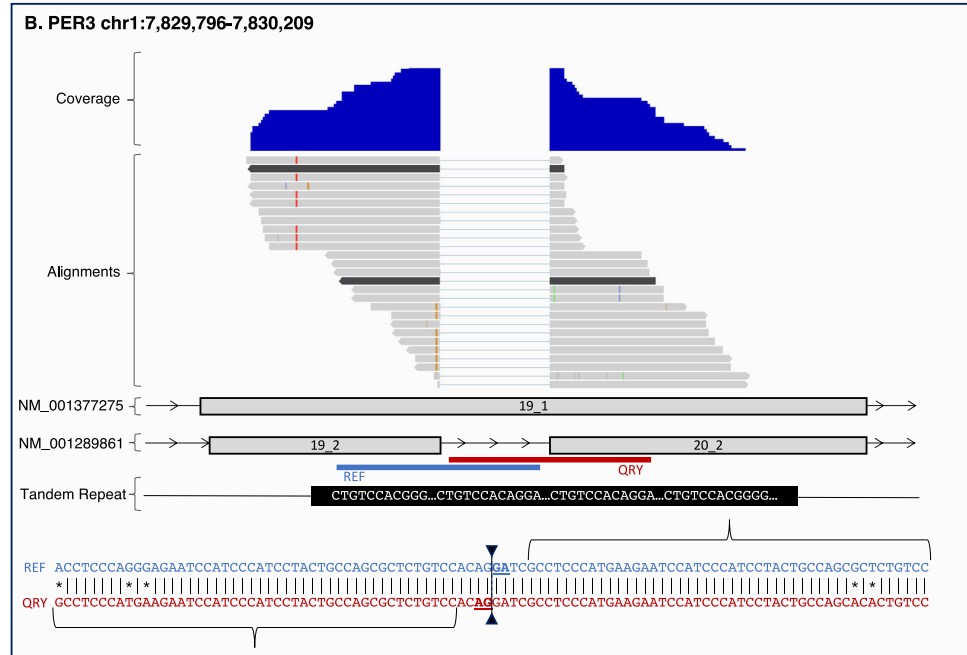

**Fig. 4 | Examples of splicing errors in reference annotation transcripts caused by complex repetitive structures or polymorphisms. A** shows an error in *NBPF20* transcript NM_001278267 where a complex repetitive structure causes an intron to be included between exons 125_1 and 126_1, skipping exon 125_2. The correct form of the transcript occurs in MANE (NM_001397211) and includes the missed exon as well as longer versions of each of the flanking exons. The repeat that causes the error includes the 100 bp alignment between the upstream (REF) and downstream (QRY) flanking regions shown below the transcripts. Inverted triangles at the center of the 100 bp alignment mark the splice sites. Above the transcripts in the figure, two tracks are displayed: the coverage track and the alignment track of spliced reads across 23 brain (DLPFC) samples, which were combined into a single alignment file using TieBrush[31]. In the coverage and alignment tracks, only the alignments that support the junction between exons 124_1 and 125_1 are shown. **B** displays an error in *PER3* transcript NM_001289861 caused by a tandem repeat. Exon 19_1 in MANE transcript NM_001377275 overlaps with a 54 bp tandem repeat, and an intron of the same length is erroneously inserted between exons 19_2 and 20_2 in NM_001289861. The first 46 bp of QRY and the last 46 bp of REF (as shown by the braces), represent the genomic region overlap between the flanking sequences. Inverted triangles at the center of the alignment denote the splice sites. The coverage and alignment of spliced reads (alignment track) across 23 DLPFC samples further support the presence of the error.

splicing events: EASTR flagged 221 instances in GENCODE, 20 in CHESS, and 10 in RefSeq. Additionally, we identified an instance where the protein-coding gene *NPIPB3* is entirely absent from the MANE catalog, likely due to differences between RefSeq and GENCODE regarding the "correct" splice site within a VNTR region in the final last exon (Supplementary Note 3).

**Zea mays**
We evaluated the performance of EASTR on six RNA-seq datasets from *Zea mays*, consisting of three biological replicates each from mature pollen and lower leaves affected by gray leaf spot disease. Previous research has shown higher TE expression in reproductive tissue

compared to vegetative tissue[11], leading us to hypothesize that the mature pollen dataset would contain a larger number of spurious alignments. Our results supported this hypothesis, indicating that EASTR was more effective in identifying false alignments in the mature pollen dataset than in the lower leaf dataset.

With respect to alignment accuracy, our results indicated that EASTR identified a higher proportion of spurious alignments in the mature pollen dataset compared to the lower leaf dataset (Fig. 3). In the mature pollen dataset, EASTR flagged 12.3% (1,840,959/14,923,699) and 14.8% (1,548,226/10,467,851) of HISAT2 and STAR spliced alignments as spurious, respectively, while only 0.8% (120,215/ 15,177,186) and 0.4% (61,286/13,981,211) were flagged in the leaf dataset. In the

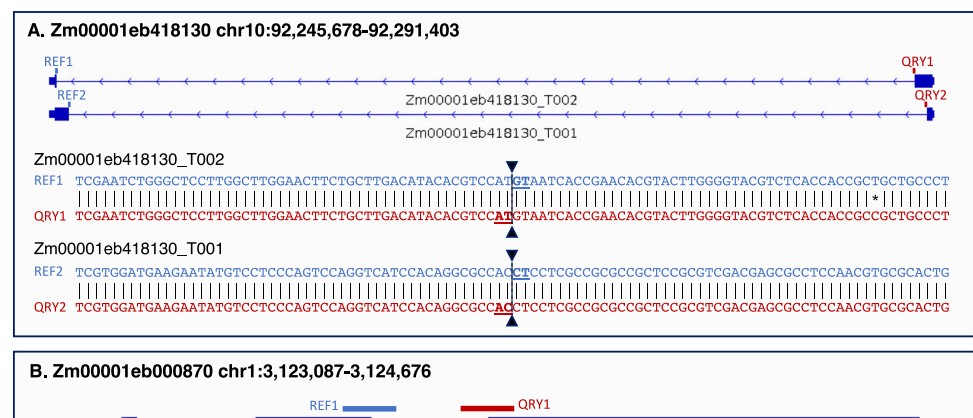

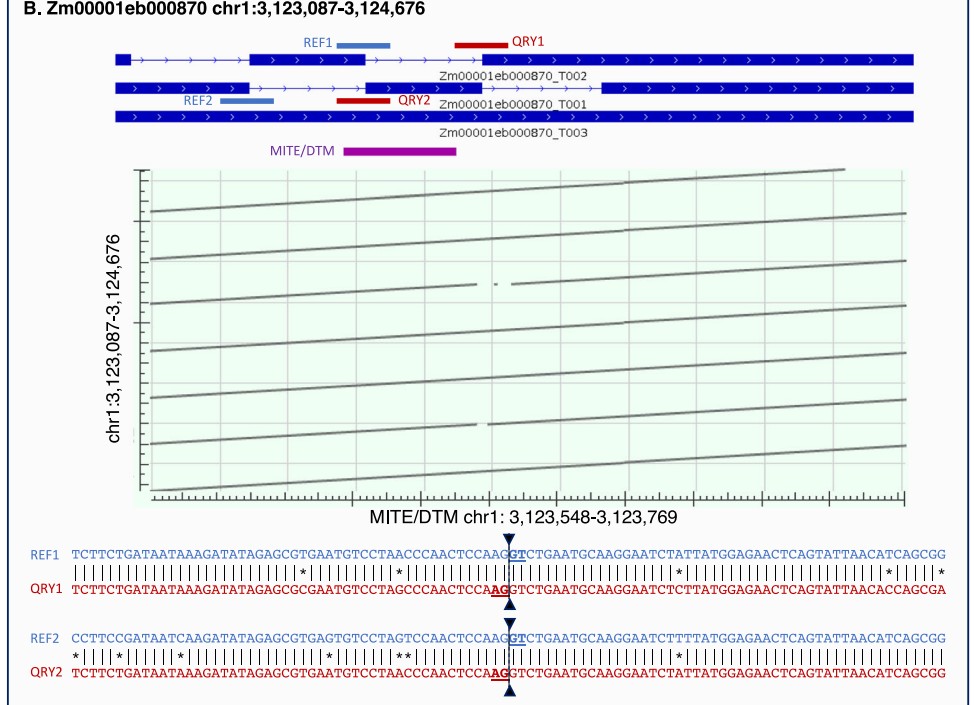

**Fig. 5 | Erroneous splice site annotation between duplicated sequences.**
**A** illustrates an instance of an error in splice site annotation involving two duplicate genes. The reference annotation (panel A, top) presents these genes as a single gene with two spliced transcripts. Alignments of the upstream and downstream intron-flanking sequences (REF1, REF2 vs. QRY1, QRY2, respectively) in transcripts Zm00001eb418130_T002 and Zm00001eb418130_T001 show near perfect conservation. **B** presents a case of splice site annotation error between duplicated transposable elements (TEs). The top track displays three annotated transcripts,

with transcript T003 being a fragment and not a full-length transcript. The TE annotation track below the transcripts shows that only a single TE is annotated in this region. However, the dot plot below the TE annotation track indicates that the annotated TE is repeated 7 times in the chr1:3,123,087–3,124,676 region. The 100 bp intron-flanking sequence alignments of two junctions in transcripts T002 and T001 are shown below the dot plot and demonstrate strong homology between the four sequences.

pollen dataset, only a small proportion of removed HISAT2 and STAR alignments, (1.0% and 1.2%, respectively) corresponded to junctions in the reference annotation (79 for HISAT2 and 87 for STAR). The remaining 13,338 and 6645 alignments for HISAT2 and STAR corresponded to non-reference junctions. In the leaf dataset, 8.8% and 13.6%, respectively of removed HISAT2 and STAR alignments supported reference junctions, corresponding to 260 and 234 annotated splice sites, with the remaining alignments supporting non-reference junctions (5538 and 1786, respectively). More details are provided in Supplementary Data 3 and Fig. 3.

Building on these findings, our analysis with StringTie2 revealed that EASTR's alignment filtering significantly enhanced transcriptome assembly accuracy, as detailed in Table 1. In the pollen dataset, the number of non-reference introns, exons, and transcripts was reduced, without compromising transcript-level sensitivity or the number of reference-matching transcripts assembled (Supplementary Data 6 and 9). Filtering the leaf dataset with EASTR also resulted in a decrease in

the number of non-reference introns, exons, and transcripts. While EASTR improved transcript-level precision in the leaf samples, there was a slight reduction in sensitivity (0.2–0.3% for HISAT2 and 0.2% for STAR) and in the number of reference-matching transcripts assembled (<40 removed out of >15,000 total). Taken together, these results suggest that filtering spurious alignments can result in a more precise and comparably sensitive transcriptome assembly.

Concluding our examination of the *Z. mays* datasets, we applied EASTR to evaluate the maize genome annotation obtained from MaizeGDB[3,22] (version 5.0 of the B73 inbred line). This analysis flagged 412 potentially spurious introns within 539 transcripts and 261 genes (Supplementary Data 14). Our analyses revealed that tandemly repeated sequences were the main cause of erroneous splice site annotation, as illustrated in Fig. 5. In Fig. 5A we show the annotation of a single gene with two spliced transcripts at the chr10:92,245,678–92,291,403 locus. However, analysis with EASTR suggests that these two transcripts of gene *acco3* may represent tandemly duplicated genes rather

than two transcripts of the same gene. Both transcripts are 1320 bp long and share 99.4% sequence identity. In Fig. 5B, we illustrate a case of likely splice site mis-annotation between duplicated TEs. While the B73 TE annotation track displays a single TE in this region, the dot plot reveals that the TE is present in seven tandem repetitions within the region. EASTR identified four potentially spurious introns annotated within this tandem repeat region.

### *A. thaliana*

We evaluated EASTR's performance on paired RNA-seq datasets from wild-type (WT) and DNA methylation-free mutant *A. thaliana* plants. The mutant plants (*mddcc*) were generated by knocking out all DNA methyltransferases (MET1, DRM1, DRM2, CMT3, and CMT2), which play an important role in maintaining DNA methylation patterns and regulating gene expression, including the silencing of transposable elements (TEs)[12]. All datasets were generated using ribo-minus library preparation and consisted of three biological replicates for each condition. Our findings supported our hypothesis that the loss of DNA methylation leads to increased TE expression levels and a higher proportion of spurious spliced alignment events detected by EASTR.

Similar to our human and *Z. mays* analyses, we applied EASTR to the *A. thaliana* datasets to identify potentially erroneous spliced alignments. EASTR flagged between 0.1% to 1.4% of spliced alignments in the assembled *A. thaliana* RNA-seq data as erroneous (Supplementary Data 2). The vast majority of these alignments (94.4% from HISAT2 and 96.9% from STAR) contained junctions not supported by the reference annotation. The remaining alignments contained junctions that were found in the annotation (49 and 117, for HISAT2 and STAR, respectively). The proportion of erroneous alignments varied across samples, with the *mddcc* mutant having over a fourfold increase in the proportion of erroneous alignments compared to the wild-type strain. In the *mddcc* mutant strain, EASTR flagged 138,999/32,614,412 (0.4%) erroneous alignments in HISAT2 and 257,794/18,652,175 (1.4%) in STAR, compared to 0.1% (31,249/ 33,230,121) and 0.2% (38,682/ 15,488,709) for the wild-type data. EASTR flagged a higher number of non-reference junctions in the *mddcc* alignments (5,734 in HISAT2 and 8,410 in STAR) than in the wild-type alignments (1,535 in HISAT2 and 1,425 in STAR). These observations were consistent across all samples (Fig. 3, Supplementary Data 2).

Building on the alignment findings, we examined how EASTR filtering influenced transcriptome assembly quality. Just as with humans and *Z. mays*, the application of EASTR filtering to alignments in the *A. thaliana* dataset improved the quality of transcriptome assembly, as shown in Table 1. We assembled transcripts with StringTie2 using both unfiltered and EASTR-filtered HISAT2 and STAR alignments, and compared the assemblies to the TAIR10.1 reference annotation (Supplementary Data 8). We observed a reduction in the number of non-annotated introns, exons, and transcripts per sample (Supplementary Data 5). The incorporation of EASTR had a minimal impact on the number of reference-matching transcripts per assembly, with a marginal loss of ≤20 out of >15,000 total reference-matching transcripts per assembly for both HISAT2 and STAR (Supplementary Data 5). These observations were consistent in wild-type and mutant datasets.

Lastly, we applied EASTR to scrutinize the TAIR10.1 reference annotation for *A. thaliana*, aiming to identify potential inaccuracies. Our analysis revealed 283 introns within 316 transcripts and 193 genes that were likely to be spurious (Supplementary Data 15). Mirroring our findings in *Z. mays*, we also identified instances of splicing between putative tandem gene duplications ((Supplementary Note 4, Supplementary Fig. 4) and potentially unannotated TEs. Our analysis also uncovered numerous annotation errors in several repeat-rich gene families, such as the Receptor-like proteins (RLP) with Leucine-rich repeat (LRR) domains[23]. This family encompasses 57 members, and we identified numerous spurious introns within this gene family (*RLP18*, *RLP34*, and *RLP49*). Within the polyubiquitin family, which contains tandem repeats of 228 bp encoding a ubiquitin monomer, we identified annotation errors in *UBQ4*, *UBQ10*, *UBQ11*, and *UBQ14*[24].

## Discussion

EASTR is a new computational tool that effectively identifies incorrect spliced alignments caused by repeat elements in RNA-seq datasets. By utilizing sequence similarity between the downstream and upstream sequences flanking a given splice junction, EASTR can identify and remove spuriously spliced alignments and also highlight potential errors in genome annotation, thereby improving the accuracy of downstream analyses that rely on alignment and annotation data.

In this study, we analyzed RNA-seq data from three species representing different tissue types and library preparation methods. Our analysis revealed that spurious alignments can account for up to 20% of the spliced alignments in the datasets we examined. The ribo-minus library preparation method had a higher proportion of spurious junctions and alignments compared to poly(A) selection, possibly because it captures nascent transcripts containing intronic sequences, which are typically enriched for repeat elements. In some samples, as many as 99.97% of the alignments that EASTR flagged for removal were not found in the reference annotation, suggesting they were likely spurious. Additionally, we observed a stark contrast in spurious alignments of reads sequenced from germline and somatic tissues of *Z. mays*, likely due to the different levels of TE expression in these two tissue types. Our findings underscore the importance of considering library preparation methods and tissue types when interpreting spliced alignment results and demonstrate EASTR's broad applicability in improving the accuracy of RNA-seq data alignment.

Our experiments also show that pre-filtering RNA-seq alignment files with EASTR can improve the accuracy of transcriptome assembly. We observed an increase in transcript assembly precision and a reduction in the count of novel (non-annotated) exons, introns, and transcripts in all samples. As detailed in Supplementary Note 5, these improvements cannot be replicated by simply adjusting the aligners' parameters. Furthermore, as outlined in Supplementary Note 6, spurious junctions are often shared across multiple samples. This points to a systematic trend in erroneous alignments, highlighting the limitations of simple threshold-based filtering approaches and underscoring the need for more sophisticated methodologies.

Our EASTR-based analysis of reference gene catalogs illustrates how past errors in spliced alignment might have produced erroneous annotation that remains in these databases today. In all gene catalogs we examined, we found hundreds of likely cases of mis-annotation. One notable finding involved a transcript in the high-quality MANE human gene catalog containing an intron flanked by two *Alu* elements, an unlikely event requiring two consecutive exonization events (Supplementary Note 1). Accurate transcript annotation remains a challenge across all eukaryotic species, and the errors we observed here are likely to be repeated in many other genome annotation databases, in which EASTR has the potential to identify similar problems.

In conclusion, EASTR offers an effective solution for detecting spurious spliced alignments and annotation errors, and can substantially improve the accuracy of RNA-seq data alignment, transcript assembly, and annotation across diverse organisms and sequencing datasets. Nonetheless, achieving precise transcript annotation remains challenging, particularly in species characterized by active transposons, high genomic TE composition, and frequent tandem gene duplication, underscoring the need for continued development of tools and methods to tackle this challenge.

## Methods

Splice-aware aligners may incorrectly map a read originating from a contiguous repeat element as a spliced alignment, especially if there

are mismatches between the read and the reference genome. For example, if a read originating from a repeat sequence contains a mismatch to the reference genome, the aligner may attempt to optimize the alignment score by splicing it to a nearby, similar, repeat sequence (Fig. 6). Mismatches between RNA-seq reads and the reference genome are relatively common and can occur for various reasons, including variation between the individual and the reference genome, RNA editing events, and sequencing errors. A spurious spliced alignment of a read that originates from a repeat sequence can manifest in two ways: (1) the first part of the read is correctly aligned to one copy of the repeat, and the second part is spliced to another similar repeat nearby (Fig. 6A); or (2) the read comes from a repeat in one locus but has multiple mismatches compared to the reference genome, and the aligner finds a higher-scoring spliced alignment between two similar repeats in a different locus (Fig. 6B).

### EASTR algorithm for detecting and removing spurious spliced alignments

EASTR aims to resolve the issue of erroneous splicing between repeat elements by recognizing sequence similarity between the flanking upstream and downstream regions of a specific intron, and the frequency of flanking sequence occurrence in the reference genome. The tool accepts various input data types and formats, including alignments from splice-aware aligners such as HISAT2 or STAR (BAM files), and reference annotations (GTF files). Details on the runtime and memory requirements for different input options can be found in Supplementary Note 7. The comprehensive workflow for detecting and removing spurious spliced alignments is described below and detailed in Fig. 2.

1. **Identification of potential repeat-induced spliced alignments**
   The input to EASTR is a set of alignments produced by a spliced aligner such as HISAT2 or STAR. For every intron identified in the input file, EASTR computes an alignment to discern similarity between an upstream "reference" sequence (centered on the 5' splice site) and a downstream "query" sequence (centered on the 3' splice site). EASTR utilizes the `mappy` Python wrapper of minimap2[25] for this purpose, with k-mer length, minimizer window size, and chaining scores set to 3, 2, and 25, respectively. By default, EASTR extracts sequences centered on both ends of the splice junction, extending a fixed distance (50 bp by default)

in either direction from the splice site. The exonic portion of each sequence is called an "exon overhang". The upstream and downstream sequences, designated as the reference and query, respectively, are aligned to each other with `mappy`. If more than one good alignment is detected, EASTR selects the primary `mappy` alignment. The alignment is scored using a matrix that assigns 3 points for a match, 4 points penalty for a mismatch, 12 points penalty for opening a short gap, 32 points penalty for opening a long gap, 2 points penalty for extending a short gap, and 1 point penalty for extending a long gap. This scoring matrix is designed to be permissive in order to capture diverged homologous sequences, such as TE families, that may have sufficient homology to be erroneously splice-aligned due to stretches of exact matching bases. An analysis exploring the sensitivity of EASTR in detecting spurious splicing, by varying combinations of match score and overhang length, is detailed in Supplementary Note 8.

2. **Classification of alignments as two- or one-anchor alignments**
   EASTR examines the `mappy` alignment generated in step 1 (if present), and classifies it as either a "two-anchor alignment" or a "one-anchor alignment" (see Fig. 2, panels B and C, respectively, for specific examples of such alignments). Here, 'anchor' refers to the portion of the alignment that extends beyond the splice junction, aligning with the neighboring exonic sequence. The anchor must satisfy the aligner's minimum length requirement for the spliced alignment to be reported. Typically, aligners like STAR and HISAT2 use minimum anchor sizes of 5–7 bp for unannotated junctions; EASTR's default anchor size is 7 bp.
   Two-anchor alignments (illustrated in Fig. 7A) meet the following criteria, considering the alignment's zero-based coordinate system: 1) both reference and query alignment starting positions are less than or equal to the exon overhang (default: 50 bp) minus the anchor length, or 43 bp by default, 2) both reference and query alignment ending positions exceed the overhang plus anchor length, adjusted for the zero-based coordinate system, or 56 bp by default, and 3) the alignment shift between reference and query sequences (absolute value of reference start minus query start) is below twice the anchor length (14 bp by default). Alignments not meeting these criteria are designated as one-anchor alignments (Fig. 7B).

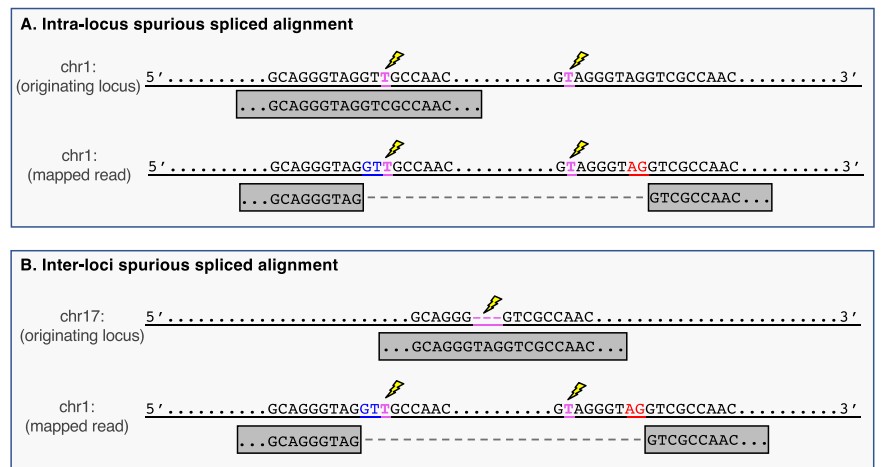

**Fig. 6 | Schematic representation of spurious spliced alignment between repeat elements. A** Intra-locus alignment error: a read (gray box) originating from an upstream repeat element on chromosome 1 has a T to C mutation (highlighted in magenta and marked by a lightning bolt) relative to the reference genome. This read also has a single mismatch to the repeated downstream sequence. An aligner might erroneously create an intron between the two repeat elements, with the canonical GT-AG splice sites highlighted in blue and red. **B** Inter-loci alignment error: a read (gray box) originating from chromosome 17 has a 3 bp insertion (highlighted in magenta and marked by a lightning bolt) relative to the reference genome. An aligner may align this read elsewhere in the genome and erroneously create an intron between two repeat elements.

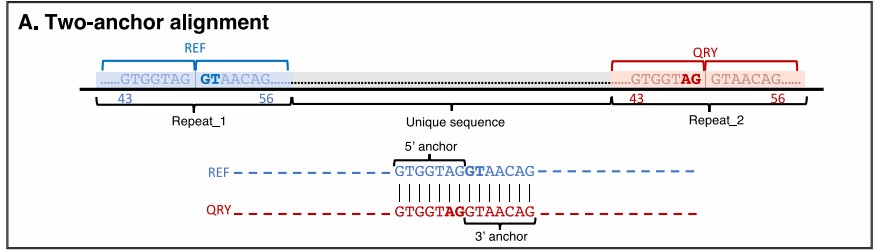

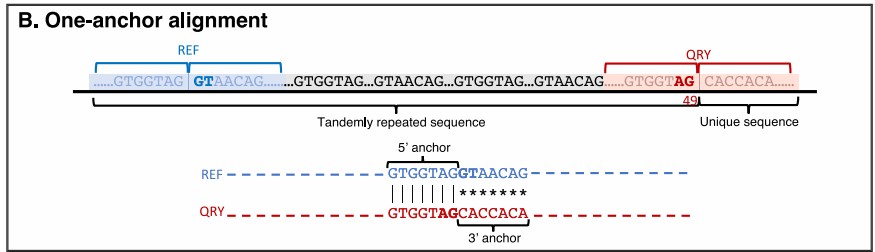

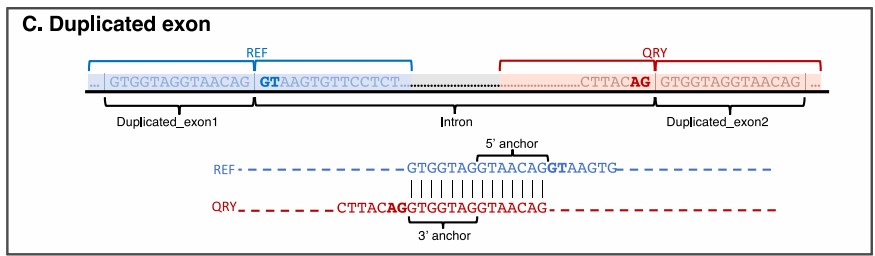

**Fig. 7 | Examples of various alignments in the EASTR workflow.** The figure highlights the different scenarios that EASTR evaluates to detect and remove spurious spliced alignments caused by sequence similarity between upstream ("REF") and downstream ("QRY") flanking regions. **A** Two-anchor alignment with similar flanking sequences; **B** One-anchor alignment with skewed similarity towards the 5' end; **C** Duplicated exon scenario where alignment between the query and the reference sequences is primarily confined to exonic regions flanking the splice sites.

3. **Bowtie2 alignment to check occurrence frequency**

   Following the initial `mappy` alignment, EASTR uses `bowtie2`[26] to map the upstream "reference", downstream "query", and "hybrid" sequences (described below) back to the reference genome. This step is essential for detecting repetitive sequences whose high occurrence increases the likelihood of them causing erroneously spliced alignments. To perform this alignment, we extract three 30 bp sequences: two from the center of the `mappy` alignment for the upstream and the downstream sequences and one hybrid sequence obtained by concatenating the 15 bp upstream of the 5' splice site with the 15 bp downstream of 3' splice site. Using bowtie2 with parameters "`-k 10 ---end-to-end -D 20 -R 5 -L 20 -N 1 -i S,1,0.50`" we map all three sequences back to the reference genome and count the number of times each aligns.

4. **Evaluation of two-anchor alignments**

   For sequence pairs classified as two-anchor alignments, EASTR uses `bowtie2` to further assess the uniqueness of the reference, query and hybrid sequences as described in step 3. If either the upstream or downstream alignment is unique and the hybrid sequence does not align elsewhere, the spliced alignment is deemed non-spurious. Conversely, if `mappy` finds an alignment and `bowtie2` aligns both the reference and query sequences to more than one genomic location, EASTR marks the splice junction as spurious.

5. **Evaluation of one-anchor alignments**

   One-anchor alignment between the reference and query flanking sequences could suggest a spurious spliced alignment. For instance, such cases can occur when there is sequence similarity between the 5' ends of the reference and query sequences, but not between the 3' ends (i.e., the repeat element causing the issue is not centered on the splice junction and is skewed toward the 5' end on both ends of the splice junction, as illustrated in Fig. 7B). In such situations, EASTR employs a two-step approach to address these scenarios:

   a. **Identifying duplicated exons:**

   EASTR initially determines whether an alignment corresponds to a duplicated exon, which is not considered a spurious spliced alignment. In these cases, the alignment between the query and the reference sequences is primarily confined to exonic regions flanking the splice sites, leading to a shifted alignment of the query and reference sequences (Fig. 7C). EASTR examines whether the query start site is shifted by distance of ≥43 bp relative to the reference start site, with 43 bp being the default threshold. This shift threshold corresponds to EASTR's minimum length for recognizing duplicate exons. If such a shift occurs and the hybrid sequence is not present elsewhere in the genome, the alignment likely represents a pair of duplicated exons rather than a spurious junction.

   b. **Identifying spurious one-anchor alignments:**

   If the alignment does not meet the criteria for a duplicated exon, EASTR examines whether the upstream and downstream partially aligned sequences appear more than 10 times in the reference genome. If this is the case, the partial alignment is deemed spurious.

**Reference genomes and annotations**

Human reads were aligned to the GRCh38 genome assembly (RefSeq accession GCF_000001405.39) after excluding pseudoautosomal regions and alternative scaffolds. The accuracy of the human

transcriptome assemblies generated using StringTie2, as well as the non-reference and reference junction counts, were evaluated by comparing them to the GRCh38.p8 release of the RefSeq annotation, filtered to include only full-length protein-coding and long non-coding RNA transcripts. The filtered RefSeq annotation can be downloaded from the following FTP link: ftp://ftp.ccb.jhu.edu/pub/EASTR. *A. thaliana* reads were aligned to TAIR10.1 (RefSeq accession GCF_000001735.4), and the accuracy of the transcriptome assemblies and junction counts were assessed by comparing them to the corresponding annotation[27]. *Z. mays* reads were aligned to the B73 NAM 5.0 assembly (RefSeq accession GCF_902167145.1) and the accuracies of the transcriptome assembly and junction counts were evaluated by comparing them to the corresponding NAM 5.0 Zm00001eb.1 annotation obtained from MaizeGDB[22] [https://www.maizegdb.org/]. The transposon annotation for *Z. Mays* was also retrieved from MaizeGDB [https://www.maizegdb.org/].

### Alignment and assembly

A HISAT2 index was built using the following command: `hisat2-build -p 16 --exon genome.exon --ss genome.ss genome.fa hisat_index`. All RNA-seq datasets were aligned using HISAT2 with default parameters using the following command: `hisat2 -x hisat_index -1 R1.fastq -2 R2.fastq -S aligned.sam`. For *Arabidopsis* and maize lower leaf datasets, we added the `--rna-strandedness RF` flag to indicate an fr-firststrand library. Sorting and converting the resulting SAM files to BAM format was done with samtools (version 1.13)[28].

A STAR index was built using: `STAR --runThreadN 12 --runMode genomeGenerate --genomeDir star_index --genomeFastaFiles genome.fa --sjdbOverhang [read_length-1] --sjdbGTFfile reference.gtf`. RNA-seq datasets were aligned and sorted by STAR using the following command: `STAR --runThreadN 12 --genomeDir star_index --readFilesIn R1.fastq R2.fastq --outSAMstrandField intronMotif --twopassMode Basic --outSAMtype BAM SortedByCoordinate --limitBAMsortRAM 16000000000 --outSAMunmapped Within --outFileNamePrefix sampleID`.

Transcriptome assembly was performed using StringTie2 (version 2.0), utilizing HISAT2 (version 2.2.1) and STAR (version 2.7.8a_2021-03-08) alignments, with the following command: `stringtie2 aligned.bam -o sample.gtf`.

### Assembly accuracy metrics

Sensitivity was quantified as the ratio of true positives (TP) to the sum of true positive and false negative (FN) exons, introns, or transcripts that match the reference annotation. Precision was quantified as the ratio of TP to the sum of TP and false positives (FP). True positives were defined as exons, introns, transcripts, or loci that match the reference annotation, and false negatives as exons, introns, transcripts, or loci in the annotation that were missing from the assembly. We used `gffcompare`[29] to count TP, FN, and FP as well as to profile the sensitivity and precision at the exon, intron, and transcript levels.

To assess the impact of EASTR on the number of novel (non-annotated) and reference-matching introns, exons, and transcripts in each StringTie2 assembly, we calculated the percent change using the formula:

$$\frac{x_f - x_i}{x_i} \times 100\% \tag{1}$$

where $x_i$ represents the count of introns, exons, or transcripts before applying EASTR to alignment files and $x_f$ represents the count after assembly using EASTR-filtered alignments. We employed the same percent change metric to evaluate the relative changes in transcript-level precision and sensitivity.

### Reporting summary

Further information on research design is available in the Nature Portfolio Reporting Summary linked to this article.

### Data availability

All RNA-seq datasets analyzed in this study are publicly available from the NCBI Sequence Read Archive. The human DLPFC dataset is available from the NCBI Sequence Read Archive under accession number PRJNA595606. The maize leaf dataset is available under accessions SRR10095075, SRR10095076, SRR10095077. The maize pollen dataset is available under accession numbers SRR3091548, SRR3091717, SRR3094513. The *A. thaliana* dataset is available under accessions SRR14056780, SRR14056781, SRR14056782, SRR16596898, SRR16596890, SRR16596900. Transcriptome assemblies generated in this study are available for download from the following FTP link: ftp://ftp.ccb.jhu.edu/pub/EASTR. There are no restrictions on data use or controlled access conditions.

### Code availability

The EASTR code used in this study is publicly available and has been archived on GitHub with a DOI identifier: https://doi.org/10.5281/zenodo.530774518[30]. The EASTR code and the analyses used in this paper are available through GitHub at https://github.com/ishinder/EASTR and https://github.com/ishinder/EASTR_analyses, respectively.

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

## Acknowledgements
We thank Steven Salzberg for his helpful feedback and insightful comments on this manuscript. This work was supported in part by NSF grant DBI-1759518 (M.P.) and NIH grant RO1-MH123567 (M.P.).

## Author contributions
I.S. conceived the study, designed and implemented the software, conducted computational analyses, analyzed and interpreted the results, and wrote the manuscript. R.H. analyzed the results. H.J.J. and K-H.C. aided in the development of the software. M.P. conceived the study, advised the design and implementation of the software, and contributed to writing and editing the manuscript. All authors reviewed and approved the final manuscript.

## Competing interests
The authors declare no competing interests.
