## [Peer Review File · Nature Communications]

EASTR: Identifying and eliminating systematic alignment errors in multi-exon genesREVIEWER COMMENTS

Reviewer #1 (Remarks to the Author):

The manuscript “The EASTR: Correcting systematic alignment errors in multi-exon genes.” By Shinder et al. describes software designed to remove alignments that are likely artifactual related to challenges of performing spliced read alignment around repetitive regions. The manuscript is well-written and the software looks to be generally useful. The method is shown to be applicable to both HISAT and STAR RNA-seq alignments, can eliminate substantial numbers and fractions of suspect RNA-seq alignments, and the results are shown to improve on transcriptome assembly. The authors’ also demonstrate that it can be used to examine existing reference genome annotations for species and identify likely artifacts in that context as well.

Major critiques:

This work with EASTR is reminiscent of much earlier work to correct spliced alignment errors around micro-exons (PMID: 12799353). However, although the title of the EASTR manuscript indicates ‘correcting’, it would appear that it is instead eliminating altogether the suspicious alignments. If EASTR is not correcting the alignments and instead filtering or removing them, then the title should better reflect this. I would note that Filtering Alignments of Spliced Transcript Reads would change the software name to FASTR.

It is clear how alignment software might make mistakes around repetitive sequences, but it’s not always entirely obvious which alternative alignment is the correct one, due to sequencing error and not a priori knowing where exactly the transcript was derived. While it is clear that removing suspect alignments around known reference gene structure annotations, which are naturally deficient in repetitive sequences, can improve on transcript reconstruction for those genes, I worry that studies of novel transcripts around repetitive regions such as HERVs might suffer, which could impact studies into expression and transcript reconstruction at or near endogenous retroviruses.

However, this issue could be explored using simulated data, so as to examine the accuracy of EASTR – both the overall accuracy, and the local genomic contextual accuracy (ex. Around coding genes, vs. around lncRNAs or HERVs).

I would suggest the following experiment:

- Start with an operational truth set for transcript structures that includes both transcribed repetitive sequences and known gene sequences. This could be done by using StringTie transcripts from the standard pipeline (not using EASTR). These isoform structures would serve as the 'operational truth set'.
- Simulate reads for those 'truth set' isoforms based on their expression values from that sample and taking into account the sequencing error profiles from corresponding samples. This is readily done using the RSEM Simulator.
- Align these simulated reads to the genome using HISAT and STAR, and determine what the alignment error is.
- Run EASTR and examine which alignments get filtered, and examine accuracy of the filtering in known genic regions and intergenic regions.

Another issue that could be explored is the effect of the alignment parameters on having incorrect alignments around repetitive sequences. Part of the issue that's responsible for this is that the aligners are likely augmenting scores of spliced alignments that have breakpoints at consensus splice sites. The authors could reduce the augmentation score (usually a command line parameter) to a minimal value and see if this also improves on alignment accuracy – easily seen with the above simulation experiment. It would be interesting to see if this reduces the need for further downstream EASTR application.

The terms 'one-way' and 'two-way' anchor are confusing, and I suspect Figure 6 (pt2) has the terms swapped. Perhaps there's different terminology that could be used that's easier to tie to the concept, like 'two-anchor' and 'one-anchor' without the 'way'.

For determining whether the splice junction regions are repetitive, it isn't clear whether the repetitiveness definition is restricted to the local region where the alignment is derived, or if it's across the whole genome. If the read is repetitive across the whole genome, it may be that the genome placement for that read is unclear and so the alignment might have lower mapping quality. But having a read aligned to the correct location but misaligned at that location is a different concept, and for repetitiveness to impact that scenario, the read region would need to be locally repetitive (separately from being globally repetitive). It isn't clear from the current read repetitiveness definition whether it reflects local vs. global, and whether this concept is relevant to EASTR as one might expect. The authors might comment on this issue of global vs. local repetitiveness and if it is considered by EASTR.

Minor critiques:

The current manuscript flow involves describing the impact of removing spurious junctions, the effect on transcriptome assembly, and applications to reference annotations to identify suspicious transcript structures. This is applied systematically to human, Maize, and Arabidopsis data. Instead of organizing the manuscript according to organism, since the main focus is the software itself and its application, perhaps the sections should be organized according to the functionality: removing spurious junctions, impact on transcriptome assembly, and then evaluating annotation accuracy – with subsections according to organism. This might be easier to read and more focused. Otherwise, it's more like rehashing the same things over again but with different organisms. I leave this to the authors' discretion.

Another potential use case for EASTR in correcting alignments could involve those cases where diverged RNA-seq samples are aligned to a related reference genome. Polymorphisms between strains or related species would make for more challenging alignments, especially around repetitive regions, and EASTR might find a suitable use case here.

Reviewer #2 (Remarks to the Author):

This paper describes EASTR, software which addresses the problem of incorrect alignments of reads to a genome due to repeats. The analysis in the paper shows that this is an important problem in genome annotation and the analysis of RNA-seq data, even in species (like Arabidopsis) with relatively few repeats. It is well-written and describes useful software that looks like it should be widely adopted (I didn't try it).

The results section addresses the applications of EASTR to three species: humans, Zea Mays and Arabidopsis. The program itself is described only in methods. I don't think this is appropriate for the paper that introduces EASTR. At a minimum, there should be a paragraph at the beginning of the results section summarizing the description of Easter that is currently in methods, including what is now Fig. 6, with some mention of each of the five steps that are described. Fig. 6 itself could be improved by making it easier to understand without working back and forth to the text (e.g. the labels 5'SS, 3'SS, query and reference could be added to panel 1). The "one-way" and "two-way" anchor alignment labels are apparently switched in panel 2. A larger figure would be justified.

There are features of the software itself that should be in this paper. For example, what is the input and what is the output. Also, there are a number of arguments that are presented in the paper as fixed,

when the github page shows that they can be adjusted. It's likely that the optimal values for some of these parameters (particularly, the overhang length and mappy parameters) will differ according to read length and the repeat structure of the genome being analyzed. I would guess that the authors explored this a bit and could provided guidance.

Reviewer #3 (Remarks to the Author):

The authors present a method for the identification and removal of spurious spliced alignments and spurious splice junctions from BAM and annotation files, respectively. The authors have identified an interesting corner case not properly handled by commonly used spliced aligners and demonstrate how the resulting false alignments translate into downstream analyses and even into reference annotations. Specifically, the corner case entails an improper handling of alignments that involve anchors to repeat-sequences in the reference. Using a pairwise re-alignment of the junction-flanking regions, the suggested approach re-evaluates the validity of splice-junctions and reports potentially spurious ones as output.

Overall the manuscript is well written and clearly lays out both the problem as well as the suggested solution. The presented approach is evaluated on alignments of three different species (human, *Zea mays*, and *Arabidopsis thaliana*) demonstrating good generalisability of the approach. In all three different use cases, the authors find spurious junctions and show that their software effectively detects them. In addition, the authors find a correlation between complexity of the genome sequence (in terms of repeats and expressed repeat elements) and the number of detected spurious junctions.

I think the presented work is an important addition to the portfolio of post-alignment analyses. While it would be worthwhile to incorporate the suggested filtering directly into the aligner-logic to not report the spurious alignments in the first place, this can be considered as out-of-scope for the presented work.

My feedback is overall quite positive and I would only like to make the following minor suggestions:

* Figure 2: The individual x-axes are normalized to the extremes of the respective signal. As the absolute ranges between plots differ quite a bit, this makes it hard to see the relative differences between the individual species / samples. (e.g., comparing SRR10689483 and SRR10689482, the relative changes for HISAT2 are roughly -7% and -10%, respectively, but visually the latter looks even smaller than the former).

* As the effect is sequence specific and thus systematic, I would assume that filtering junctions by redundancy across samples (as is commonly done in studies) would not help to address the spurious

junctions detected by EASTR. Do the authors in fact find spurious junctions shared across multiple samples of the same species? If yes, what is the fraction of shared vs individual? How would this look for larger cohorts (e.g., GTEx) - are there junctions that are commonly found but are actually not there?

* The effect of EASTR is measured as percentage of spliced alignments detected as spurious (e.g., in the order of 3% for human). How would that fraction look when made unique over junctions? What fraction of junctions reported by the aligner is spurious?

* Currently neither the main manuscript nor the supplement contain any mention of how expensive it is to run EASTR. Would be good to get an estimate how heavy this post-filtering weighs compared to the alignment.

* In line 57f the authors write: "HISAT2 [8] addresses this by prioritizing spliced alignments over contiguous alignments when mapping quality is similar ...". What does "similar" mean in this context? Specifically, I was wondering about the example given in Figure 5A. In the current example, the downstream repeat sort of compensates for the mismatch at the original location. However, depending on how strictly the above prioritization of the spliced alignment is, this case might even occur if the downstream repeat instance would not compensate for the mismatch but generate a (generally preferred) spliced alignment. This would make this problem much more severe.

* In the description of the EASTR algorithm the language could be improved to be more precise:

** In step 1, it would be good to mention that the authors consider all viable local alignments between query and reference above a certain threshold. In the beginning, I was wondering where the different alignments come from until I realized that they not only pick the best one.

** In step 2 in line 383 the authors write: "...sequence that must align at the ends of a splice junction...". Please improve the wording to make this more precise.

** In step 2, line 389 the authors write about the "exon overhang". Please define what this is. It might in general be helpful to either augment Figure 7 to also contain these concepts (exon overhang, anchor, ...) or introduce a separate figure to explain the concept.

** In step 2, relating to the comment above. The authors mention positions a lot, where I believe they refer to positions using the coordinate system of the alignment in question. Here also it would be helpful to have this either reflected in Figure 7 or an alternative figure.

** In step 4, lines 414-415 the authors write: "aligns both sequences to more than one genomic location". What does "both" refer to in this context? Also, are there any locality constraints for genomic region or would it also be considered spurious if alignments to a different chromosome occur?

* In Figure 6, panel 3, the authors explain the selection of sequences for Bowtie2 realignment. In the text (lines 402-404), they write "we extract three 30bp sequences: two from the center of the mappy

alignment for the upstream and the downstream sequences and one hybrid sequence obtained by concatenating the ...". I have a hard time mapping this to what is shown in the figure. By definition, both reference and query should have the same length and center around the junction. However, in the figure the junction is not in the center and it seems for reference and query the same sequence is selected. What do the dashed vs solid parts represent? It would be great to have more clarity here.

* Figure 7 - I think the caption would read better if the last sentence became the first one, as it is not specific to any of the panels, but describes the figure as a whole.

* Line 457 - there is a misspelling of MaizeGDB

* In the section "Alignment and assembly" (lines 459 and following), for some tools there is only a "[ref]" placeholder instead of the actual reference. Also, some tools in this section lack versions, e.g., HISAT2 or STAR.

* In Supplemental section 1, the authors use splice site predictions from SpliceAI as additional evidence to corroborate their findings. As the SpliceAI input window is usually quite large (as correctly found by the authors) at least the local repeat structure of the genome should be picked up by the ML model. Did the authors check whether it generally holds true that the junctions that EASTR reports as spurious also have a lower score for at least one of their acceptor or donor signals according to SpliceAI?

Reviewer #1 (Remarks to the Author):

The manuscript "The EASTR: Correcting systematic alignment errors in multi-exon genes." By Shinder et al. describes software designed to remove alignments that are likely artifactual related to challenges of performing spliced read alignment around repetitive regions. The manuscript is well-written and the software looks to The method is shown to be applicable to both HISAT and STAR RNA-seq alignments, can eliminate substantial numbers and fractions of suspect RNA-seq alignments, and the results are shown to improve on transcriptome assembly. The authors' also demonstrate that it can be used to examine existing reference genome annotations for species and identify likely artifacts in that context as well.

Major critiques:

This work with EASTR is reminiscent of much earlier work to correct spliced alignment errors around micro-exons (PMID: 12799353). However, although the title of the EASTR manuscript indicates 'correcting', it would appear that it is instead eliminating altogether the suspicious alignments. If EASTR is not correcting the alignments and instead filtering or removing them, then the title should better reflect this. I would note that Filtering Alignments of Spliced Transcript Reads would change the software name to FASTR.

We thank the reviewer for drawing our attention to previous work related to correcting spliced alignments around micro-exons and for the thoughtful remark regarding the terminology in our manuscript's title. We chose the term 'correcting' based on our view that by filtering out suspicious alignments, EASTR effectively 'corrects' the alignment file, leading to better overall mapping accuracy. EASTR can also be used to correct a given annotation file by removing erroneous transcripts that were incorrectly assembled due to the presence of such suspicious alignments. Nevertheless, we understand the reviewer's suggestion to more directly reflect the 'filtering' process in the title. As a result, we have amended the title to "EASTR: Identifying and eliminating systematic alignment errors in multi-exon genes," while maintaining the tool's name as EASTR to align with the aforementioned rationale.

It is clear how alignment software might make mistakes around repetitive sequences, but it's not always entirely obvious which alternative alignment is the correct one, due to sequencing error and not a priori knowing where exactly the transcript was derived. While it is clear that removing suspect alignments around known reference gene structure annotations, which are naturally deficient in repetitive sequences, can improve on transcript reconstruction for those genes, I worry that studies of novel transcripts around repetitive regions such as HERVs might suffer, which could impact studies into expression and transcript reconstruction at or near endogenous retroviruses.

However, this issue could be explored using simulated data, so as to examine the accuracy of EASTR – both the overall accuracy, and the local genomic contextual accuracy (ex. Around coding genes, vs. around lncRNAs or HERVs).

I would suggest the following experiment:

- Start with an operational truth set for transcript structures that includes both transcribed repetitive sequences and known gene sequences. This could be done by using StringTie transcripts from the standard pipeline (not using EASTR). These isoform structures would serve as the 'operational truth set'.
- Simulate reads for those 'truth set' isoforms based on their expression values from that sample and taking into account the sequencing error profiles from corresponding samples. This is readily done using the RSEM Simulator.
- Align these simulated reads to the genome using HISAT and STAR, and determine what the alignment error is.
- Run EASTR and examine which alignments get filtered, and examine accuracy of the filtering in known genic regions and intergenic regions.

We thank the reviewer for appreciating the improvement potential of EASTR on studies involving novel transcripts and for drawing our attention to potential problems in assembling transcripts in or near repetitive regions, particularly HERVs. Recognizing the validity of this observation, we conducted additional experiments to demonstrate EASTR's ability to effectively differentiate between real and spurious spliced alignments. Even for novel transcripts that are not near or within repetitive sequences, it is challenging to precisely discern their real intron-exon structure. Using simulated data from a reference annotation is usually used to evaluate how well an assembly tool does to recognize this structure. However, we have no way to know the real splicing around the repetitive elements, and we are cautious about employing simulated reads from StringTie assemblies that could be wrong themselves since they are based on erroneous spliced alignments (which EASTR is designed to identify) as a definitive "truth set."

Instead, we used SpliceAI - the current state-of-the-art method in splice site prediction, which was shown to be highly accurate in scoring splice junctions from an arbitrary pre-mRNA transcript sequence – to score splice site junctions overlapping HERV elements. Our hypothesis was that EASTR's effectiveness in distinguishing spurious from genuine junctions would lead to higher SpliceAI scores among retained junctions. Indeed, our findings aligned with this expectation. Specifically, we encountered 375 HERV-to-HERV junctions, identified by either HISAT or STAR, that were removed by EASTR. In contrast, 804 HERV-to-HERV junctions identified by either aligner were preserved. The SpliceAI scores for EASTR-filtered junctions were notably lower, ranging from 6 to 19 times lower, compared to the scores for the retained

ones. We present these results in Section 1 of the Supplemental material, and added the following paragraph to the 'Results' section of the paper:

“Identifying spurious splicing events within transcribed repetitive regions, such as Human Endogenous Retroviruses (HERVs), presents a complex challenge due to the abundance of highly similar copies that often lead to multiple alignments or spuriously spliced alignments. However, not all spliced alignments in these regions are erroneous, emphasizing the need for EASTR to carefully discern between valid and spurious splice sites. To assess EASTR's robustness in differentiating between genuine and spurious junctions in this context, we used SpliceAI¹⁷, a machine learning based splice site prediction tool, to score splice junctions that overlap HERV elements in the previously analyzed 23 DLPFC samples. We hypothesized that EASTR would predominantly retain junctions with higher SpliceAI scores, filtering out those with notably lower scores. Of the 1,179 HERV-to-HERV junctions identified by either HISAT or STAR, EASTR removed 375 junctions. Consistent with our hypothesis, these removed junctions had SpliceAI scores 6 to 19 times lower than the scores of the retained junctions (further details are provided in Supplemental Materials Section 1).”

Another issue that could be explored is the effect of the alignment parameters on having incorrect alignments around repetitive sequences. Part of the issue that's responsible for this is that the aligners are likely augmenting scores of spliced alignments that have breakpoints at consensus splice sites. The authors could reduce the augmentation score (usually a command line parameter) to a minimal value and see if this also improves on alignment accuracy – easily seen with the above simulation experiment. It would be interesting to see if this reduces the need for further downstream EASTR application.

Following the reviewer's suggestion, we conducted experiments to see if increasing the penalty for spliced alignments could potentially enhance alignment accuracy. With this in mind, we tested two scenarios: one using HISAT2's default canonical splice penalty of zero (“No penalty”), and another applying a canonical splicing penalty of 12 (“With penalty”), which matches HISAT's default non-canonical splice penalty.

Our findings, presented in Tables S1 and S2 of Supplemental Materials, showed that the "with penalty" approach lowered the number of predicted novel junctions, while also consistently resulted in fewer matches to reference junctions across all samples. However, the reduction in novel junctions (Table S1) did not lead to an improvement in transcript-level sensitivity or precision (Table S2). Instead, the method that proved most effective in increasing both sensitivity and precision of transcript assembly was utilizing HISAT2 without a canonical splice penalty, coupled with EASTR filtering.

The detailed procedures and results of this investigation can be found in Section 5 of the Supplemental Material. Additionally, in the Discussion paragraph where we state that we “observed an increase in transcript assembly precision and a reduction in the count of novel

(non-annotated) exons, introns, and transcripts in all samples” we have added the following sentence:

“As detailed in Supplemental Materials Section 5, these improvements cannot be replicated by simply adjusting the aligners’ parameters. “

The terms ‘one-way’ and ‘two-way’ anchor are confusing, and I suspect Figure 6 (pt2) has the terms swapped. Perhaps there’s different terminology that could be used that’s easier to tie to the concept, like ‘two-anchor’ and ‘one-anchor’ without the ‘way’.

Thank you for pointing out the confusion with the terms 'one-way' and 'two-way' anchor in Figure 6. Upon review, we found that the terms were indeed swapped in the figure, and we have now corrected this error. We also redesigned Figure 6 (now updated as Figure 2 in the revised manuscript) to improve its comprehensibility. We also concur with the reviewer’s suggestion to enhance clarity by using the terminology 'two-anchor' and 'one-anchor,' omitting the word 'way.' Accordingly, we have made these changes throughout the manuscript to improve readability.

For determining whether the splice junction regions are repetitive, it isn’t clear whether the repetitiveness definition is restricted to the local region where the alignment is derived, or if it’s across the whole genome. If the read is repetitive across the whole genome, it may be that the genome placement for that read is unclear and so the alignment might have lower mapping quality. But having a read aligned to the correct location but misaligned at that location is a different concept, and for repetitiveness to impact that scenario, the read region would need to be locally repetitive (separately from being globally repetitive). It isn’t clear from the current read repetitiveness definition whether it reflects local vs. global, and whether this concept is relevant to EASTR as one might expect. The authors might comment on this issue of global vs. local repetitiveness and if it is considered by EASTR.

EASTR's approach involves aligning sequences flanking a given splice junction to assess their similarity, which could stem from both "locally repetitive" regions, such as tandem repeats, and "globally repetitive" elements like *Alu* elements found throughout the genome. EASTR does not explicitly differentiate between these “local” and “global” scenarios. Instead, EASTR's primary differentiation lies in the categorization of alignments as "one-anchor" or "two-anchor." Two-anchor alignments imply that there is sufficient alignment between the two flanking regions that an aligner may splice align a given read originating from either end.

A one-anchor alignment might indicate a situation where the repeat is confined to the exonic regions of the flanking sequences, possibly resulting from a duplicated exon. In cases where the duplicated sequence appears frequently in the genome, EASTR considers this in its evaluation. As detailed in the Methods section, an additional layer of complexity is addressed by examining the uniqueness of the concatenated sequence formed by joining the 5' exon to the 3' exon.

EASTR ensures that a contiguous alignment of this region is not present elsewhere in the genome, thereby ruling out the likelihood of it being a "global" repeat.

Additionally, a one-anchor alignment might occur due to variable number of tandem repeat (VNTR) polymorphisms between the reference genome and an individual sample. An illustrative example of this scenario is depicted in Figure 7B, where a 3' unique region flanks a VNTR. In such an instance, the downstream 3' exonic portion (the unique region) would not align, whereas the downstream 3' intronic region would align to the 5' exonic region.

We have incorporated two paragraphs at the beginning of the results section to summarize EASTR's workflow, emphasizing these key aspects. We believe that this addition will provide readers with a more comprehensive understanding of the nuanced approach EASTR employs in discerning spliced alignments and addressing the complex scenarios described by the reviewer.

Minor critiques:

The current manuscript flow involves describing the impact of removing spurious junctions, the effect on transcriptome assembly, and applications to reference annotations to identify suspicious transcript structures. This is applied systematically to human, Maize, and Arabidopsis data. Instead of organizing the manuscript according to organism, since the main focus is the software itself and its application, perhaps the sections should be organized according to the functionality: removing spurious junctions, impact on transcriptome assembly, and then evaluating annotation accuracy – with subsections according to organism. This might be easier to read and more focused. Otherwise, it's more like rehashing the same things over again but with different organisms. I leave this to the authors' discretion.

We appreciate the reviewer's suggestion to reorganize the manuscript according to functionality, with subsections dedicated to different organisms. We actually initially considered this particular organization as well, but we decided to go with our current structure which allows for a species-centric focus. This enables readers who are primarily studying a specific genome or species to find all the pertinent information in one place, facilitating a seamless reading experience for those who want to understand how EASTR performs across different functionalities for a particular organism. Additionally, each species has a unique experimental design, including different library types, developmental stages, germline or somatic tissues, and methylation patterns, that impact alignment and transcriptome assembly differently. By focusing on species and experimental design, we are able to point out these nuances. We believe that the current organization aligns well with the objective of the manuscript, maintaining a balance between general principles and species-specific insights. Therefore, we would prefer to retain the current structure.

Another potential use case for EASTR in correcting alignments could involve those cases where diverged RNA-seq samples are aligned to a related reference genome. Polymorphisms between strains or related species would make for more challenging alignments, especially around repetitive regions, and EASTR might find a suitable use case here.

We appreciate the reviewer's suggestion of another interesting application for EASTR, specifically for correcting alignments where diverged RNA-seq samples are aligned to a related reference genome. Indeed, polymorphisms between strains or related species can complicate alignments, especially in repetitive regions, and this is an area where EASTR could be particularly beneficial. We actually conducted an experiment where we aligned the 23 human samples we used in the manuscript to the Pan Troglodytes genome. While these results were favorable towards the application of EASTR, we decided not to include them as they would require a much more substantial analysis to assess the accuracy of all the results, going beyond the scope of the current manuscript.

Reviewer #2 (Remarks to the Author):

This paper describes EASTR, software which addresses the problem of incorrect alignments of reads to a genome due to repeats. The analysis in the paper shows that this is an important problem in genome annotation and the analysis of RNA-seq data, even in species (like Arabidopsis) with relatively few repeats. It is well-written and describes useful software that looks like it should be widely adopted (I didn't try it).

The results section addresses the applications of EASTR to three species: humans, Zea Mays and Arabidopsis. The program itself is described only in methods. I don't think this is appropriate for the paper that introduces EASTR. At a minimum, there should be a paragraph at the beginning of the results section summarizing the description of Easter that is currently in methods, including what is now Fig. 6, with some mention of each of the five steps that are described. Fig. 6 itself could be improved by making it easier to understand without working back and forth to the text (e.g. the labels 5'SS, 3'SS, query and reference could be added to panel 1). The "one-way" and "two-way" anchor alignment labels are apparently switched in panel 2. A larger figure would be justified.

This is an insightful observation and suggestion. We agree that introducing EASTR within the Results section is essential, and we have now included a two-paragraph summary of EASTR's methodology at the beginning of the Results section together with an updated version of Figure 6, now Figure 2 in the revised manuscript.

Regarding Figure 6 (now Figure 2), we have followed the reviewer's suggestions for improvement by incorporating the specified labels (5' SS, 3' SS) to enhance comprehension. The entire figure was redesigned in order to improve clarity and only essential elements were kept in order not to distract from the main points of the algorithm.

There are features of the software itself that should be in this paper. For example, what is the input and what is the output. Also, there are a number of arguments that are presented in the paper as fixed, when the github page shows that they can be adjusted. It's likely that the optimal values for some of these parameters (particularly, the overhang length and mappy parameters) will differ according to read length and the repeat structure of the genome being analyzed. I would guess that the authors explored this a bit and could provided guidance.

We appreciate the reviewer's keen observation regarding the potential variability in optimal values for certain parameters, such as overhang length and match score, which may be influenced by read length and the specific repeat structure of the genome under analysis. Recognizing the importance of this insight, we have introduced a new section in the supplemental material, 'Section 7: EASTR Parameter Selection.' This section elaborates on the rationale for our default settings and offers an in-depth examination of our exploration into how these parameters might impact the results. We made sure to specify throughout the manuscript that these values are not fixed and only represent default values. We also added a new sentence in the main manuscript to refer the reader to this section:

“An analysis exploring the sensitivity of EASTR in detecting spurious splicing, by varying combinations of match score and overhang length, is detailed in Supplemental Materials Section 7.”

Furthermore, to provide a clear understanding of EASTR's functionality, we have included the following detailed description of the software's input and output in the Methods section:

“The tool accepts inputs in various data types and formats, including alignments from splice-aware aligners like HISAT2 or STAR (BAM files), as well as reference annotations (GTF files). Details on the runtime and memory requirements for different input options can be found in Supplemental Materials Section 8.”

Supplemental Materials Section 8 outlines the various data types and formats EASTR can accommodate and provides additional information concerning the runtime and memory requirements for different input options.

Reviewer #3 (Remarks to the Author):

The authors present a method for the identification and removal of spurious spliced alignments and spurious splice junctions from BAM and annotation files, respectively. The authors have identified an interesting corner case not properly handled by commonly used spliced aligners and demonstrate how the resulting false alignments translate into downstream analyses and even into reference annotations. Specifically, the corner case entails an improper handling of alignments that involve anchors to repeat-sequences in the reference. Using a pairwise re-alignment of the junction-flanking regions, the suggested approach re-evaluates the validity of

splice-junctions and reports potentially spurious ones as output.

Overall the manuscript is well written and clearly lays out both the problem as well as the suggested solution. The presented approach is evaluated on alignments of three different species (human, *Zea mays*, and *Arabidopsis thaliana*) demonstrating good generalisability of the approach. In all three different use cases, the authors find spurious junctions and show that their software effectively detects them. In addition, the authors find a correlation between complexity of the genome sequence (in terms of repeats and expressed repeat elements) and the number of detected spurious junctions.

I think the presented work is an important addition to the portfolio of post-alignment analyses. While it would be worthwhile to incorporate the suggested filtering directly into the aligner-logic to not report the spurious alignments in the first place, this can be considered as out-of-scope for the presented work.

The reviewer is right that integrating our filtering directly into the aligners' logic would be the best approach. While this is indeed out-of-scope for our paper, we intentionally designed a separate tool to ensure flexibility across many different aligners and we thank the reviewer for appreciating the value of our work.

My feedback is overall quite positive and I would only like to make the following minor suggestions:

* Figure 2: The individual x-axes are normalized to the extremes of the respective signal. As the absolute ranges between plots differ quite a bit, this makes it hard to see the relative differences between the individual species / samples. (e.g., comparing SRR10689483 and SRR10689482, the relative changes for HISAT2 are roughly -7% and -10%, respectively, but visually the latter looks even smaller than the former).

The reviewer's observation about the inconsistency in the visual representation of the relative changes between individual species/samples is well-noted and valuable. We acknowledge that the normalized x-axes in Figure 2 (now Figure 3), tailored to the extremes of each respective signal, may have obscured the true comparative differences.

In response, we have adjusted the scaling of the x-axes across all plots based on the number of non-reference junctions in the HISAT2 alignment. Specifically, the width of the x-axis from the lower to the upper limit is now set to 60% of the HISAT2 x-value before applying EASTR.

This adjustment ensures a consistent visual representation of relative changes. A relative change of 10% from "HISAT2" to "HISAT2 with EASTR" now corresponds to the same distance across all species/samples, represented as 1/6 of the x-axis. Similarly, a 20% change appears twice as distant (1/3 of the x-axis). This uniform scaling facilitates a more intuitive and accurate comparison of the relative changes across the various species and samples in the study.

* As the effect is sequence specific and thus systematic, I would assume that filtering junctions by redundancy across samples (as is commonly done in studies) would not help to address the spurious junctions detected by EASTR. Do the authors in fact find spurious junctions shared across multiple samples of the same species? If yes, what is the fraction of shared vs individual? How would this look for larger cohorts (e.g., GTEx) - are there junctions that are commonly found but are actually not there?

The reviewer raises an important question regarding whether spurious junctions detected by EASTR are shared across multiple samples or unique to individual ones, and how this might look for larger cohorts such as GTEx. In response to this inquiry, we conducted an analysis using 489 heart tissues from GTEx, focusing on understanding how spurious junctions are distributed across samples of the same species.

Our findings indeed reveal a systematic pattern in spurious alignments, with a significant portion of these being shared across samples. Specifically, we observed that 76% of spurious junctions supported by 5 or more alignments are shared across 5 or more individuals within the 489 GTEx samples.

These insights are detailed in Supplemental Materials Section 6, which includes Figure S5. Figure S5 in the supplemental material illustrates the count of shared spurious junctions across different numbers of samples. Two insets in Figure S5 provide detailed views of specific ranges, highlighting the distribution of junctions shared by both smaller and larger numbers of samples. We are now drawing the reader's attention to this supplemental text with the following addition to the main text:

“Furthermore, as outlined in Supplemental Materials Section 6, spurious junctions are often shared across multiple samples. This points to a systematic trend in erroneous alignments, highlighting the limitations of simple threshold-based filtering approaches and underscoring the need for more sophisticated methodologies.”

* The effect of EASTR is measured as percentage of spliced alignments detected as spurious (e.g., in the order of 3% for human). How would that fraction look when made unique over junctions? What fraction of junctions reported by the aligner is spurious?

The fraction of spurious junctions among those reported by the aligner is detailed in Supplemental Tables 1.1 to 1.3, where we have added a column to present the percentage of removed junctions (made unique over all junctions).

* Currently neither the main manuscript nor the supplement contain any mention of how expensive it is to run EASTR. Would be good to get an estimate how heavy this post-filtering

weighs compared to the alignment.

The reviewer pointed out the importance of understanding the computational expenses related to running EASTR and how this post-filtering process weighs in comparison to the alignment. We have added "Section 8: EASTR Runtime and Memory Requirements" to the Supplemental Materials to provide a detailed analysis of the time and resources needed to run EASTR, and refer the reader to it in the main text, as follows:

“The tool accepts inputs in various data types and formats, including alignments from splice-aware aligners like HISAT2 or STAR (BAM files), as well as reference annotations (GTF files). Details on the runtime and memory requirements for different input options can be found in Supplemental Materials Section 8.”

To summarize, EASTR accepts various input formats, such as individual BAM files, GTF annotation files, and batch processing of a list of BAM files. When processing 23 individual HISAT2-aligned human BAM files, the runtime ranged from 3.2 to 46.8 CPU minutes per file, with a consistent maximum resident set size (RSS) of 3.2 GB. For human RefSeq annotation, EASTR required approximately 1 minute and 32 seconds of wall clock time, and for batch processing of 46 BAM files, the wall clock time was around 53 minutes and 17 seconds, with a maximum RSS of 4.3 GB.

These metrics demonstrate that EASTR's computational requirements are reasonable and compatible with typical alignment processing pipelines. The efficiency can be further enhanced by utilizing a list of BAM files to avoid redundant checks of shared junctions among samples.

* In line 57f the authors write: "HISAT2 [8] addresses this by prioritizing spliced alignments over contiguous alignments when mapping quality is similar ...". What does "similar" mean in this context? Specifically, I was wondering about the example given in Figure 5A. In the current example, the downstream repeat sort of compensates for the mismatch at the original location. However, depending on how strictly the above prioritization of the spliced alignment is, this case might even occur if the downstream repeat instance would not compensate for the mismatch but generate a (generally preferred) spliced alignment. This would make this problem much more severe.

In the context of HISAT2, the term "similar" referred to a scenario where the spliced alignment has no mismatches in comparison to the contiguous alignment, or when both alignments are considered equally good. In such cases, HISAT2 prioritizes the spliced alignment or reports both alignments, depending on the exact circumstances.

We have edited this line to read: “HISAT2 [8] addresses this by prioritizing spliced alignments over contiguous alignments if the spliced alignment has no mismatches compared to the

contiguous alignment (Figure 1). If the two alignments are equally good, HISAT2 will report both alignments.”

The reviewer is correct in their observation about the potential severity of this issue, particularly if the downstream repeat instance does not compensate for a mismatch. In such a scenario, where the contiguous alignment and the spliced alignment have the same number of mismatches and are deemed "equally as good" (as stated in the 2015 HISAT paper), HISAT will report both the contiguous and spliced alignments. This behavior indeed exacerbates the problem.

* In the description of the EASTR algorithm the language could be improved to be more precise:

** In step 1, it would be good to mention that the authors consider all viable local alignments between query and reference above a certain threshold. In the beginning, I was wondering where the different alignments come from until I realized that they not only pick the best one.

We thank the reviewer for pointing out that we weren't clear about how EASTR treats viable local alignments. When mappy identifies more than one possible alignment, EASTR selects the primary alignment as designated by mappy. We say this explicitly in Step 1 now, and we also modified Figure 6 (now Figure 2) to clarify how EASTR works.

** In step 2 in line 383 the authors write: "...sequence that must align at the ends of a splice junction...". Please improve the wording to make this more precise.

Based on the reviewer's feedback, we have revised this section of step 2 to provide a clear definition of what anchors mean in the context of our method. Here's the addition we made:

" Here, 'anchor' refers to the portion of the alignment that extends beyond the splice junction, aligning with the neighboring exonic sequence. The anchor must satisfy the aligner's minimum length requirement for the spliced alignment to be reported."

** In step 2, line 389 the authors write about the "exon overhang". Please define what this is. It might in general be helpful to either augment Figure 7 to also contain these concepts (exon overhang, anchor, ...) or introduce a separate figure to explain the concept.

Based on the reviewer's feedback, we have added a sentence to Step 1 to explicitly define what the exon overhang means in the context of our method. The addition is as follows:

“By default, EASTR extracts sequences centered on both ends of the splice junction, extending a fixed distance (50bp by default) in either direction from the splice site. The exonic portion of each sequence is called an “exon overhang”.”

We have also updated Figure 6 (now Figure 2) to clearly mark the exon overhangs in the figure, which are now explained in the figure's caption:

“Deeper hues of blue and red, referred to as exon overhangs (default: 50bp), denote segments originating from “exonic” regions.”

** In step 2, relating to the comment above. The authors mention positions a lot, where I believe they refer to positions using the coordinate system of the alignment in question. Here also it would be helpful to have this either reflected in Figure 7 or an alternative figure.

We recognize the potential for confusion around the coordinate system used in the description of two-anchor alignments. In response, we have rewritten Step 2 to explicitly state that the criteria for two-anchor alignments are defined with respect to the alignment's zero-based coordinate system. We have edited the last paragraph in step 2 as follows (changes are highlighted):

“Two-anchor alignments (illustrated in Figure 7A) meet the following criteria, considering the alignment’s zero-based coordinate system: 1) both reference and query alignment starting positions are less than or equal to the exon overhang (default: 50bp) minus the anchor length, or 43bp by default, 2) both reference and query alignment ending positions exceed the overhang plus anchor length, adjusted for the zero-based coordinate system, or 56bp by default, and 3) the alignment shift between reference and query sequences (absolute value of reference start minus query start) is below twice the anchor length (14bp by default). Alignments not meeting these criteria are designated as one-anchor alignments (Figure 7B).”

** In step 4, lines 414-415 the authors write: "aligns both sequences to more than one genomic location". What does "both" refer to in this context? Also, are there any locality constraints for genomic region or would it also be considered spurious if alignments to a different chromosome occur?

The reviewer has rightly identified an ambiguity in our original description. In this context, "both" refers to the reference and query sequences adjacent to the splice junction. To clarify, we have modified the text to explicitly mention "both the reference and query sequences," thus removing any confusion.

Regarding locality constraints, EASTR's evaluation of spurious alignments does not focus on specific chromosomes or genomic regions. Please also see our answer to the first reviewer’s comment about locally versus globally repetitive elements.

* In Figure 6, panel 3, the authors explain the selection of sequences for Bowtie2 realignment. In the text (lines 402-404), they write "we extract three 30bp sequences: two from the center of the mappy alignment for the upstream and the downstream sequences and one hybrid sequence obtained by concatenating the ...". I have a hard time mapping this to what is shown

in the figure. By definition, both reference and query should have the same length and center around the junction. However, in the figure the junction is not in the center and it seems for reference and query the same sequence is selected. What do the dashed vs solid parts represent? It would be great to have more clarity here.

We appreciate the reviewer's careful examination of Figure 6, panel 3, and recognize that the representation might have been unclear. The confusion might arise from the way we are looking at the alignment of reference and query sequences. We are only considering the portion of these two sequences that align, and that portion may not be symmetric around the junction, as they could be shifted relative to one another. In response, we have completely redesigned Figure 6 (now Figure 2) to better illustrate how EASTR algorithm works, and we now specifically indicate this central region in the figure, while also explaining it in the caption:

“the alignment is deemed spurious if a defined central region (default:30bp, marked by a gray square) within the aligned flanking sequences maps to multiple genomic locations as identified by bowtie2”

* Figure 7 - I think the caption would read better if the last sentence became the first one, as it is not specific to any of the panels, but describes the figure as a whole.

We agree with reviewer's suggestion and moved the last sentence in the Figure 7's caption to the beginning of the caption.

* Line 457 - there is a misspelling of MaizeGDB

We thank the reviewer for bringing this to our attention. We have corrected the misspelling of 'MaizeGDB' (we have also identified a second instance of this misspelling and have corrected it).

* In the section "Alignment and assembly" (lines 459 and following), for some tools there is only a "[ref]" placeholder instead of the actual reference. Also, some tools in this section lack versions, e.g., HISAT2 or STAR.

We have added version numbers and updated the “[ref]” placeholder with appropriate references.

* In Supplemental section 1, the authors use splice site predictions from SpliceAI as additional evidence to corroborate their findings. As the SpliceAI input window is usually quite large (as correctly found by the authors) at least the local repeat structure of the genome should be picked up by the ML model. Did the authors check whether it generally holds true that the junctions that EASTR reports as spurious also have a lower score for at least one of their acceptor or donor signals according to SpliceAI?

We appreciate the reviewer's insightful question regarding the relationship between EASTR's identified spurious junctions and SpliceAI scores. To address this query (as well as another question from the first reviewer), we have conducted an additional analysis, specifically focusing on Human Endogenous Retroviruses (HERVs) within the human genome.

The reason for selecting HERV-to-HERV junctions is that HERV elements may undergo splicing, and not all junctions within these regions should be removed. Since HERV elements can exploit splicing to diversify their compact genomes, it is important to ensure that EASTR properly discriminates between genuine and spurious splice sites, thus avoiding over-filtering of valid splicing events.

In the newly added Section 5 of the Supplemental Materials, titled "Validation of EASTR filtering within transcribed repetitive genomic regions," we examined the junctions overlapping HERV elements in 23 DLPCF human RNA-seq samples. We utilized SpliceAI to score all splice junctions and compared the results for junctions removed and retained by EASTR.

The analysis revealed that junctions identified by EASTR as spurious indeed had lower average SpliceAI scores for both acceptor and donor signals (0.05 and 0.009, respectively), while the retained junctions exhibited significantly higher average scores (0.3 for acceptors and 0.17 for donors). Furthermore, none of the removed junctions had either splice donor or splice acceptor scores above 0.6, contrasting with 37 such instances among the retained junctions.

We present these results in Section 1 of the Supplemental material, and added the following paragraph to the 'Results' section of the paper:

"Identifying spurious splicing events within transcribed repetitive regions, such as Human Endogenous Retroviruses (HERVs), presents a complex challenge due to the abundance of highly similar copies that often lead to multiple alignments or spuriously spliced alignments. However, not all spliced alignments in these regions are erroneous, emphasizing the need for EASTR to carefully discern between valid and spurious splice sites. To assess EASTR's robustness in differentiating between genuine and spurious junctions in this context, we used SpliceAI¹⁷, a machine learning based splice site prediction tool, to score splice junctions that overlap HERV elements in the previously analyzed 23 DLPCF samples. We hypothesized that EASTR would predominantly retain junctions with higher SpliceAI scores, filtering out those with notably lower scores. Of the 1,179 HERV-to-HERV junctions identified by either HISAT or STAR, EASTR removed 375 junctions. Consistent with our hypothesis, these removed junctions had SpliceAI scores 6 to 19 times lower than the scores of the retained junctions (further details are provided in Supplemental Materials Section 1)."

REVIEWERS' COMMENTS

Reviewer #1 (Remarks to the Author):

I thank the authors for their thoughtful responses to my critique, and I appreciate the modifications to the manuscript. My only remaining comment is that ideally EASTR would actually correct alignments where possible rather than eliminate them. Perhaps that may be a feature of a future version of the software.

Reviewer #2 (Remarks to the Author):

The revised manuscript is acceptable for publication without further revision.

Reviewer #3 (Remarks to the Author):

The authors have addressed all my comments and questions. I appreciate the additional experiments conducted on the redundancy of spurious junctions and the comparison to the Splice-AI predictions.

Reviewer #1 (Remarks to the Author):

I thank the authors for their thoughtful responses to my critique, and I appreciate the modifications to the manuscript. My only remaining comment is that ideally EASTR would actually correct alignments where possible rather than eliminate them. Perhaps that may be a feature of a future version of the software.

We appreciate the reviewer's thoughtful and constructive feedback, which has improved the quality of our work. It's worth noting that many of the alignments that EASTR currently eliminates are multi-mapped, with contiguous multi-mapped alignments being retained. The reviewer's suggestion provides valuable direction for EASTR's future development, and we appreciate the insights.

Reviewer #2 (Remarks to the Author):

The revised manuscript is acceptable for publication without further revision.

We appreciate the reviewer's assessment that the revised manuscript is ready for publication without further revisions. We thank the reviewer for their time and constructive feedback.

Reviewer #3 (Remarks to the Author):

The authors have addressed all my comments and questions. I appreciate the additional experiments conducted on the redundancy of spurious junctions and the comparison to the Splice-AI predictions.

We appreciate the reviewer's nuanced and detailed comments, which were instrumental in refining our manuscript. The reviewer's detailed review guided our revisions and enhanced the manuscript's clarity and impact. We thank the reviewer for investing their time and expertise to improve our work.